# **The ICON-based Earth System Model for Climate**

# **Predictions and Projections (ICON XPP v1.0)**

- Wolfgang A. Müller<sup>1</sup>, Stephan Lorenz<sup>1</sup>, Trang V. Pham<sup>2</sup>, Andrea Schneidereit<sup>2</sup>, Renate
- Brokopf<sup>1</sup>, Victor Brovkin<sup>1,6</sup>, Nils Brüggemann<sup>1</sup>, Fatemeh Chegini<sup>1</sup>, Dietmar Dommenget<sup>3</sup>,
- Kristina Fröhlich<sup>2</sup>, Barbara Früh<sup>2</sup>, Veronika Gayler<sup>1</sup>, Helmuth Haak<sup>1</sup>, Stefan Hagemann<sup>4</sup>,
- Moritz Hanke<sup>5</sup>, Tatiana Ilyina<sup>6</sup>, Johann Jungclaus<sup>1</sup>, Martin Köhler<sup>2</sup>, Peter Korn<sup>1</sup>, Luis
- Kornblueh<sup>1</sup>, Clarissa A. Kroll<sup>7</sup>, Julian Krüger<sup>1</sup>, Karel Castro-Morales<sup>2</sup>, Ulrike Niemeier<sup>1</sup>,
- Holger Pohlmann<sup>1</sup>, Iuliia Polkova<sup>2</sup>, Roland Potthast<sup>2</sup>, Thomas Riddick<sup>1</sup>, Manuel Schlund<sup>8</sup>,
- Tobias Stacke<sup>1</sup>, Roland Wirth<sup>2</sup>, Dakuan Yu<sup>1</sup>, and Jochem Marotzke<sup>1</sup>
- 12 <sup>1</sup> Max-Planck Institute for Meteorology, Hamburg, Germany
- 13 <sup>2</sup> Deutscher Wetterdienst, Offenbach am Main, Germany
- 14 <sup>3</sup> ARC Centre of Excellence for Climate Extremes, Monash University, Australia
- <sup>4</sup> Institute of Coastal Systems, Helmholtz-Zentrum Hereon, Geesthacht, Germany
- <sup>5</sup> Deutsches Klimarechenzentrum, Hamburg, Germany
- 17 <sup>6</sup> University of Hamburg, Hamburg, Germany
- $^{7}$  Institute for Atmospheric and Climate Science, ETH Zürich, Zürich, Switzerland
- 19 <sup>8</sup>Deutsches Zentrum für Luft- und Raumfahrt (DLR), Institut für Physik der Atmosphäre,
- 20 Oberpfaffenhofen, Germany

10

11

21

22

23

25

24 Corresponding author: Wolfgang A. Müller, wolfgang.mueller@mpimet.mpg.de

Abstract. We develop a new Earth System model configuration framed into the ICON 28 architecture, which provides the baseline for the next generation of climate predictions and projections (hereafter ICON XPP - where XPP stands for eXtended Predictions and 29 30 Projections). ICON XPP comprises the atmospheric component of the numerical weather 31 prediction (ICON NWP), the ICON ocean and land surface components, and an ensemble-32 variational data assimilation system, all adjusted to an Earth System model for pursuing climate 33 research and operational climate forecasting. Two baseline configurations are presented: 1) a 34 160 km atmosphere and a 40 km ocean resolution, and 2) 80 km atmosphere and 20 km ocean 35 resolution. A CMIP DECK (Diagnostic, Evaluation and Characterization of Klima) 36 experimentation framework is used for a first evaluation. 37 ICON XPP depicts the basic properties of the coupled climate. The pre-industrial climate shows a top-of-atmosphere balanced radiation budget and a mean global near-surface 38 39 temperature of 13.8-14.0 °C. The ocean shows circulation strengths in the range of the observed 40 values, such as the AMOC at 16-18 Sv and the flows through the common passages. The current climate is characterized by a trend in the global mean temperature of ~1.2 °C since the 41 42 1850s, similar to reference datasets. Regionally, the hydroclimate differs greatly from observed 43 conditions. For example, the inter-tropical convergence zone (ITCZ) has a double peak and a 44 wet southern subtropical branch across the oceans. Further, the Southern Ocean sea surface 45 temperature has a strong positive mean bias with temperatures up to 5 °C higher than observations. 46 47 Dynamical processes, such as El Niño/Southern Oscillation (ENSO) performs similarly to CMIP6-like coupled models. Tropical waves and the Madden-Julian Oscillation are well 48 49 captured, and the 40-km atmospheric configuration has a spontaneous weak quasi-biennial 50 oscillation. The atmospheric dynamics in the northern extra-tropics of both configurations 51 represent well the position of the jet stream as well as the influences of the transient momentum 52 transports and their feedbacks on the jet stream. Overall, ICON XPP performs similarly to

#### 1. Introduction

projections, and climate research.

For more than a decade, the Max Planck Institute Earth System model (MPI-ESM) has been

used for climate predictions and projections and climate research. Climate predictions (here

climate models performed in CMIP6 making it a good basis for climate forecasts and

58 spanning the time range from seasons to 10 years ahead) based on MPI-ESM provide reliable 59 forecast skill (Marotzke et al., 2016) and are routinely operated by the Deutscher Wetterdienst 60 (DWD) (Fröhlich et al., 2020). Further, MPI-ESM contributed to previous phases of the 61 Coupled Model Intercomparison Project (CMIP) through various configurations (e.g., 62 Giorgetta et al., 2013; Gutjahr et al., 2019; Jungclaus et al., 2013; Mauritsen et al., 2019; Müller 63 et al., 2018). However, MPI-ESM will no longer be supported, and has been substituted by the 64 ICON (ICOsahedral Nonhydrostatic) model framework. ICON XPP - where XPP stands for 65 eXtended Predictions and Projections - is a newly developed coupled Earth System model 66 configuration based on the ICON framework, becoming the baseline for the next generation climate predictions, and provides the model platform for the contribution to the CMIP7 (Dunne 67 68 et al., 2024). Here, we present ICON XPP, from the design of the configurations to a first 69 evaluation of the Earth System state. Special attention is given to monitoring certain aspects of 70 the tropical and extra-tropical mean climate, and key modes of variability.

ICON XPP advances the achievements of previous ICON initiatives related to sub-components 72 of the Earth System model (Giorgetta et al., 2018; Korn, 2017; Korn et al., 2022; Nabel et al., 73 2020; Reick et al., 2021; Schneck et al., 2022; Zängl et al., 2015), and a fully-coupled Earth 74 System model (Jungclaus et al., 2022). Although these configurations are based on the same 75 dynamical core and code infrastructure of ICON, their sub-grid scale closure and 76 parameterization differ and depend on whether they are used for weather or climate scales. 77 Since 2020, a new modeling initiative integrates numerical weather forecast (NWP), climate 78 predictions and climate projections based on the ICON framework into a single model system 79 (Müller et al., 2025). An outcome of this initiative is ICON XPP that combines some of the 80 well-established NWP and climate model components, and synchronizes the physical 81 parameterizations among weather and climate timescales (Müller et al., 2025). ICON XPP 82 consists of the atmospheric component used for operational weather forecasts at the DWD 83 (ICON NWP), which has achieved superior quality of weather forecasting compared to 84 previous NWP model generations, as well as the ICON ocean and sea-ice model and the land 85 component JSBACH.

A central aim of ICON XPP is to substitute MPI-ESM for climate predictions, upcoming climate projections and provision of basic research on fundamental climate properties. Climate predictions with the MPI-ESM have demonstrated skill at various timescales from seasons to multiple decades. On seasonal timescales, MPI-ESM shows prediction skill for various

dominant modes of climate variability such as the El Niño/Southern Oscillation (ENSO) (Fröhlich et al., 2020) and the North Atlantic Oscillation (NAO) (Dobrynin et al., 2018; Dobrynin et al., 2022). MPI-ESM has also been used for the assessment of decadal climate predictions and is used to conduct operational forecasts (Hettrich et al., 2021; Marotzke et al., 2016). Decadal prediction skill in the model has been shown to arise from near-term memory in the North Atlantic Ocean heat content and from the externally-forced long-term trends (i.e., Marotzke et al., 2016 and references therein). In addition, actual predictions skill is found for key processes, such as the Quasi-Biennial Oscillation (QBO) (Pohlmann et al., 2013), storm tracks in the northern hemisphere extra-tropics (Kruschke et al., 2016; Schuster et al., 2019), and the NAO (Athanasiadis et al., 2020; Smith et al., 2020), for climate impacts, such as continental-scale surface temperature (Müller et al., 2012) and associated extremes (Borchert et al., 2019; Wallberg et al., 2025), and Earth System processes, such as the carbon uptake in the ocean (Li et al., 2016). Recently, MPI-ESM has been used to extend the prediction skill to a multi-decadal timescale (Düsterhus and Brune, 2024). A principal ambition is that ICON XPP is able to cover predictions at all timescales from months to multi-decades. Given these targets, special emphasis is put on incorporating and improving model components particularly suitable for climate predictions. Though this attempt is quite broad, first initiatives led to the inclusion of a higher-resolving stratosphere, and special attention was paid to the key properties in the tropics and the extra-tropics.

While the development and evaluation of ICON XPP for operational climate prediction and CMIP7 is still in progress, here we present its principal development lines and fundamental properties of the coupled Earth System state. We use the DECK-experimental design - which has been developed as a guideline to improve and compare among coupled Earth System models (Eyring et al., 2016). We present the basic model description and ways towards tuning the model climate, followed by an evaluation of the basic climate state, trends, and climate sensitivity in the DECK experiments.

In ICON XPP, we paid special attention to fast and flexible model configurations, to perform long integrations and large ensembles, in contrast to current high-resolution ICON model initiatives. Long-time integrations are particularly useful while testing the parameter space finding an equilibrium state of the coupled system, but also for probing the ideal setting for improving key dynamics. Large ensembles are the standard procedure in simulations of climate projections, and to assess reliability in the ensemble forecasts and eventually to improve the

- signal-to-noise ratio by adequate methodologies (Dobrynin et al., 2018; Smith et al., 2019).
- Further, large ensembles are essential for the assessment of the transient climate variability
- (Maher et al., 2019). The ICON XPP configurations presented here are designed to run several
- simulated decades per day and are suitable for the aforementioned tasks.

## 2. Model Description, Configurations, and Tuning

### 2.1 Model Components

- ICON XPP integrates Earth System components that have been established for operational
- weather forecasting and climate application, and here are plugged together for the first time. In
- the following, the components that form ICON XPP are described in more detail.
- ICON NWP

126

- The atmospheric component of ICON XPP is based on the operational configuration of ICON
- NWP (Zängl et al., 2015). In ICON NWP, the basic non-hydrostatic model equation system is
- solved on a triangular grid. The vertical grid of ICON is a terrain-following hybrid sigma height
- grid (Giorgetta et al., 2018; Leuenberger et al., 2010), with a model top at 75 km. The
- centerpiece is the dynamical core, in which the model equations are integrated forward in time,
- followed by the numerical advection schemes and physical parameterizations (for details see
- Prill et al., 2024). ICON NWP uses the physics packages from the operational regional model
- COSMO (Doms and Schättler, 2004), and from the ECMWF Integrated Forecast System
- (Zängl et al., 2015). For radiation, the ecRad scheme is used in ICON NWP (Hogan and Bozzo,
- 2018). An overview of the physical parameterizations is given in Müller et al. (2025, Table 1).
- ICON Land
- ICON XPP uses the land surface component of ICON (ICON Land). ICON Land includes the
- JSBACH land-surface model developed for predecessors of ICON XPP such as MPI-ESM
- (Reick et al., 2021; Reick et al., 2013), and other land-surface model such as TERRA (from
- the Latin for "earth"), which is implemented into the operational configuration of ICON NWP.
- JSBACH version 3 (JSBACHv3) operated as a part of MPI-ESM in both, concentration and
- emission-driven modes, and demonstrated a good performance of terrestrial carbon cycle in
- CMIP6 (Hajima et al., 2024). JSBACH version 4 (JSBACHv4) includes climate-relevant
- physical and biogeochemical processes, such as a full carbon cycle, dynamic vegetation, and

151 land-cover changes for the land use. In addition, the soil physics in JSBACHv4 are improved 152

in permafrost regions compared to JSBACHv3. The land-surface model can be used in stand-

153 alone mode, as well as in the fully coupled Earth System models (Jungclaus et al., 2022).

For ICON XPP, JSBACH is newly implemented together with its parameterization of the vertical diffusion as an implicitly coupled module of ICON NWP. As TERRA, JSBACH accounts for subgrid heterogeneity. However, in contrast to TERRA in which tiles are treated externally, JSBACH uses them internally to account for the different land-surface types and plant functional types (PFTs) as a basis for biogeochemical processes. Therefore, a new interface layer is developed between JSBACH, its vertical diffusion scheme, and the rest of the NWP parameterizations. This new interface layer results in the adjustment of code for other sub-components. For example, parts of the sea-ice thermodynamics scheme are re-

162 implemented, and the coupling to the ocean is generalized.

167

169

170

171

172

# Hydrological Discharge Model

A hydrological discharge (HD) model is used in ICON XPP to route water from the land model 166 JSBACH to the river mouths feeding into the ocean model ICON O. In ICON XPP, we can

choose between two HD model versions. One is the internal HD model integrated within

168 JSBACH. This HD model operates at the same horizontal resolution and time step as JSBACH,

maintaining coherence between land and hydrological processes. Automatic generation of HD

parameters for ICON grids based on high resolution digital elevation data (Riddick, 2021;

Riddick et al., 2018) allows HD application on any spatial resolution using none or minimal

manual adjustments. This model is used for ICON XPP in the lower-resolved configuration

(see section 2.2).

A new version of the HD model with relatively high resolution of 0.5° is externally connected 174

to JSBACH (Hagemann et al., 2023). And this is only used in high-resolution ICON XPP

configurations. The HD model is a separate model component coupled via YAC (Yet Another

Coupler) (Hanke et al., 2016) with both, the land-surface and the ocean model components. In

this setup, the HD model is coupled to the atmosphere and the ocean with daily intervals. The

land-surface scheme of ICON NWP handles surface and subsurface runoff, which are

interpolated by YAC onto the HD latitude-longitude grid. This approach offers the advantage

of being independent of the land-surface model, allowing HD to work with other models such

- as TERRA. It will also easily allow future applications using the HD model at its higher
- resolution of 1/12° (Hagemann et al., 2020), and taking advantage of ongoing developments in
- riverine transport of biogeochemical tracers (e.g., Elizalde et al., 2025).
- ICON O/Sea-Ice
- The ocean component of ICON solves the hydrostatic Boussinesq equations of large-scale
- ocean dynamics with a free surface (Korn, 2017; Korn et al., 2022). ICON O uses the same
- horizontal grid and data structures as the atmosphere. For the vertical grid, the actual model
- uses depth-based coordinates such as z or z\*-coordinates as the default option (Korn et al.,
- 2022). For ICON XPP, we use the uniformly vertical-distributed grid with the z\*-coordinate.
- Further, a newly developed sea-ice dynamics is applied which operates on the native ICON
- grid (Mehlmann et al., 2021; Mehlmann and Korn, 2021). The sea-ice dynamic is based on
- FESIM (Danilov et al., 2015). Sea-ice thermodynamic is calculated in the atmospheric part and
- uses the zero-layer model (Semtner Jr., 1976; Mironov et al., 2012). Melting potential and
- conductive heat flux are passed to the ocean component by use of the YAC coupler.
- HAMOCC
- The ocean biogeochemistry component in ICON XPP is represented by the HAMburg Ocean
- Carbon Cycle model, HAMOCC6 (Ilyina et al., 2013; Paulsen et al., 2017), featuring biology
- and inorganic carbon chemistry processes in the water column and sediment. The growth of
- bulk phytoplankton is limited by temperature and light as well as by the availability of nutrients
- including nitrate, phosphate, and iron linked by constant Redfield ratios across organic
- compartments. The growth of nitrogen-fixing cyanobacteria is parameterized analogously to
- that of the bulk phytoplankton, albeit at a lower rate and is extended by representing their
- buoyancy. Detritus is explicitly separated into opal- and calcium carbonate-producing
- phytoplankton fractions. Zooplankton growth function is limited by the grazed phytoplankton,
- mortality, and metabolic activity. The dissolved organic matter pool is shaped by the exudation
- of phytoplankton, cyanobacteria, and zooplankton. All the biogeochemical tracers are
- transported by the flow field. HAMOCC has been extensively evaluated as part of MPI-ESM
- (e.g., Li et al., 2023; Mauritsen et al., 2019; Müller et al., 2018; Nielsen et al., 2024) and
- implemented in previous configurations of the ICON-based models (Hohenegger et al., 2023;
- Jungclaus et al., 2022). Compared to its predecessors, HAMOCC in ICON XPP incorporates a
- prognostic calculation for marine aggregate sinking speeds (Maerz et al., 2020), providing an
- improved distribution of particulate organic carbon fluxes critical to the ocean biological pump.

#### 2.2 Configuration

We use the latest ICON model version (ICON release 2024.07). Two configurations have been developed, differing mainly in spatial resolutions. The first is a high-resolution configuration, intended for operational climate prediction and projections. It utilizes the atmospheric model ICON NWP with approximately 80 km horizontal grid spacing (r2b5) and 130 vertical levels (L130) (Niemeier et al., 2023). The vertical spacing of the layers increases up to a value of 500 m at an altitude of about 14 km and stays constant (500 m) until an altitude of 35 km. Above this height the vertical distance increases until the model top at 75 km altitude (Fig. 1). This configuration uses the externally calculated HD model as described above. The ocean model operates at a resolution of about 20 km (r2b7) with 72 vertical levels (L72). The integration time steps for ICON NWP and ICON O are 450 seconds and 20 minutes, respectively. The coupling interval between the atmosphere and ocean is 60 minutes. Due to its high resolution and frequent computation intervals, this configuration is computationally expensive, but a throughput of ~45 simulated years per day on 100 nodes ensures long integrations. The experiments are run on the CPU-partition of the Levante High-Performance Computing system at the Deutsche Klimarechenzentrum (DKRZ), with each node consisting of 2 CPUs and 128 cores in total. This configuration is named "80/20" hereafter to reflect the grid-scale of the atmospheric and ocean components. The second configuration is with coarser resolutions and was developed to allow more efficient simulations. In this configuration ICON NWP is run with a 160 km grid (r2b4) and 90 vertical levels (L90) and model top at 75 km, while ICON O operates on a 40 km grid (r2b6). The HD model is implemented internally to JSBACH. Additionally, the ocean model's time step is increased to 30 minutes compared to the 80/20 configuration. The coupling interval between the atmosphere and ocean is 30 minutes. This configuration is designed for running large

ensembles and long integrations and has a throughput of ~85 simulated years per day on 40 computing nodes. This configuration is referred to as "160/40" hereafter.

Figure 1: Full-level height (km) and vertical grid spacing (m) of the vertical grids of ICON XPP 160/40 and 80/20.

Two profiles are shown for each grid resolution, one starts at sea level and one starts at a height of ~ 5 km representing the grid over mountains.

#### 2.3 Tuning

The model configurations are tuned towards pre-industrial climate targets. The targets mainly consider the top-of-atmosphere (TOA) radiation balance and global-mean temperature at 2 meters (GMT), and the strength of the Atlantic meridional overturning circulation (AMOC) and sea-ice properties. The thermodynamic state of the atmosphere is mainly controlled by parameters in the convection, microphysics and cloud cover parameterization schemes. The ocean state is controlled by the horizontal and vertical diffusion, eddy parameterizations, and sea-ice parameters. A series of tailored pre-industrial control experiments are employed to find the optimal parameterization values. First, a wider range of convection, microphysics and cloud cover parameters are examined to estimate their impacts on the TOA radiation balance and GMT. Then, with the resulting subset of atmospheric and oceanic parameters the ocean-

circulation and sea-ice distributions are adjusted. With the optimized parameters a new spinup is started. The values of the optimized parameter values are shown in Table 1.

The spin-up is started from the Polar Science Center hydrographic climatology (PHC3.0) (Steele et al., 2001). TOA radiation values are well-balanced with values of 0.2 W m<sup>-2</sup> (-0.1 W m<sup>-2</sup>) for the 160/40 (80/20) configuration. A GMT of ~13.8 °C is achieved for both configurations. Figures 2a and 2b show the evolution of the radiation and GMT. The figures illustrate that the atmosphere reaches quasi-equilibrium after ~200 years, despite small trends towards lower temperatures remaining at the end of the simulations. The ocean state is also well-balanced as indicated by the AMOC at 26° N and 1000 m depth (Fig. 2c), but requires ~500-600 years to reach equilibrium. In 160/40 a small negative trend of the AMOC remains at the end of the simulation.

The tuning of the ocean biogeochemistry is carried out after the spin-up of the coupled configuration. The target is to limit drifts in the biogeochemical tracer fields and fluxes and to drive the model closer to observations. HAMOCC tracers are initialized from a tuned standalone 40 km ocean setup, which was spun up for ~1000 years in a pre-industrial climate. The HAMOCC tuning parameters were changed accounting for the ocean circulation in the coupled model. The appropriate weathering rates were updated during the simulation, to compensate for the loss of carbon and nutrients from the water column to the sediment.

Table 1: Parameter values used for tuning the ICON XPP configuration towards the pre-industrial climate targets. The table only shows parameters which values differ with respect to the ICON NWP configurations (160/40 and 80/20) and ICON O default values. The "Default" column shows values for ICON NWP and ICON O that are in the ICON release (2024.07) and the namelist document therein. The units are given in squared bracket and dimensionless otherwise.

| Parameter Values                                   | Process            | 160km/40km | 80km/20km | Default |
|----------------------------------------------------|--------------------|------------|-----------|---------|
| ICON NWP                                           |                    |            |           |         |
| Entrainment rate [m <sup>-1</sup> ] (tune_entrorg) | Convection         | 0.0021     | 0.0028    | 0.00195 |
| Cloud cover parameter (tune_box_liq_asy)           | Cloud microphysics | 3.35       | 3.6       | 2.5     |
| Turbulent diffusion (f_theta_decay)                | Vertical diffusion | 1.0        | 1.0       | 4.0     |

| ICON O                                                              |                               |                     |                                  |                                             |
|---------------------------------------------------------------------|-------------------------------|---------------------|----------------------------------|---------------------------------------------|
| TKE mixing (c_k)                                                    | Vertical diffusion            | 0.05                | 0.1                              | 0.1                                         |
| Minimum interior mixing [m <sup>2</sup> s <sup>-2</sup> ] (tke_min) | Vertical diffusion            | 1.0e-5              | 1.0e-6                           | 1.0e-6                                      |
| Biharmonic viscosity parameter [m <sup>4</sup> s <sup>-1</sup> ]    | Horizontal velocity diffusion | 3.5e12 (no scaling) | 0.027 (scaling with edge length) | /                                           |
| Gent&McWilliams [m <sup>2</sup> s <sup>-1</sup> ] (tracer_GM_kappa) | Eddy parameterization         | 400                 | 400                              | 1000<br>(corresp. to 400<br>km grid-length) |
| Redi [m <sup>2</sup> s <sup>-1</sup> ]<br>(tracer_isoneutral)       | Eddy parameterization         | 400                 | 400                              | 1000<br>(corresp. to 400<br>km grid-length) |
| Sea-ice parameter (leadclose1)                                      | Sea-ice melting               | 0.25                | 0.25                             | 0.5                                         |
| Sea-ice parameter (leadclose2)                                      | Sea- ice freezing             | 0.666               | 0.0                              | 0.0 (Hibler)                                |

Figure 2: Climate equilibrium in CTRL indicated by the evolution of (a) TOA net radiation (W m<sup>-2</sup>), (b) GMT (°C), (c) Atlantic meridional overturning circulation (Sv), and (d) the northern hemisphere sea-ice volume (km<sup>3</sup>). In each figure, 80/20 is shown in red and 160/40 is shown in blue. The vertical lines in (c) indicate the initialization dates for HIST. In (d) solid/dashed lines represent northern hemispheric winter/summer.

## 3. Mean Climate, Trends and Climate Sensitivity

#### 3.1 DECK Experiments

We perform DECK experiments, which have become a common tool for coordinating a comparable design of global climate model simulations (Eyring et al., 2016). Pre-industrial control simulations (CTRL) for each configuration are performed based on the spin-up experiments. The spin-up and CTRL experiments consist of a total length of 1000 years . Further, ensembles of experiments with historical forcing from CMIP6 (HIST) are used to analyze the present-day evolution of climate. The initial conditions for the historical experiments are based on the coupled control climate with a 50-year lag for subsequent

members. Finally, the climate sensitivity is estimated by a 1 %  $CO_2$  increase until doubling (1% $CO_2$ ) and an abrupt 4 x  $CO_2$  (4x $CO_2$ ) experiments. Table 2 gives an overview of the experiments.

**Table 2:** List of experiments, short description and number of simulated years of DECK experiments for both configurations. For HIST three ensemble members are performed for the period 1850-2014.

| Experiment List                                      | Description                                                                          | Number of simulated years |  |
|------------------------------------------------------|--------------------------------------------------------------------------------------|---------------------------|--|
| Spin-up and pre-industrial control simulation (CTRL) | Started from Levitus and external forcing only                                       | 1000                      |  |
| Historical simulation (HIST)                         | Started from CTRL with transient external forcing                                    | 1850-2014                 |  |
| 1 % increase of CO <sub>2</sub> (1%CO <sub>2</sub> ) | Atmospheric $CO_2$ concentration prescribed to increase at 1 % $yr^{-1}$             | 150                       |  |
| 4x abrupt CO <sub>2</sub> (4xCO <sub>2</sub> )       | Atmospheric CO <sub>2</sub> concentration abruptly quadrupled and then held constant | 150                       |  |

#### 3.2 Pre-industrial Control Climate

The CTRL experiments reveal bias distributions well-known in coupled climate models. Near-surface temperatures in both configurations exhibit warm biases in the upwelling region at the coastal western boundaries (Fig. 3). A cold tongue is visible in both configurations in the tropical Pacific, and a cold bias hot spot is found along the North Atlantic Current. The Southern Ocean marks an area with a very pronounced warm bias up to 5 °C (3 °C) in 160/40 (80/20), which appears relatively large compared to the CMIP6 multi-model mean (2-2.5K) (Luo et al., 2023) and the previous model generations (Müller et al., 2018, Jungclaus et al., 2022). Preliminary analysis of the sources of these biases points towards a too deep ocean mixed layer in the Weddell Sea associated with a strong vertical mixing (not shown). In addition, the atmosphere reveals a strong short-wave net radiation bias over the Southern Ocean, which is related to the presence of few clouds. The cloud bias is also found in an atmosphere-only simulation and reveals that in this area the clouds comprise too little cloud water and too much cloud ice. The standard deviation of the global errors is ~2.4 °C for 160/40 and ~1.7 °C for 80/20, which indicates a substantial effect by the resolution increase. Such a resolution effect on the mean error is also found in the MPI-ESM (Müller et al., 2018).

The sea-ice simulations reveal reasonable distributions in the northern hemisphere winter season with 2-3 m sea-ice thickness in the central Arctic and 0.1-0.2 m within the Labrador Sea (Fig. 4, shown only for 80/20, but it is similar in 160/40). During the summer seasons in the northern and southern hemisphere both configurations show only little sea-ice thickness. The sea-ice volume of 80/20 in the northern hemisphere winter seasons is about 30 x 10<sup>3</sup> km<sup>3</sup> (Fig. 2d), which is comparable with the PIOMAS arctic sea-ice volume reanalysis (30-35 x 10<sup>3</sup> km<sup>3</sup> April value during 1980s) (Zhang and Rothrock, 2003), and 13 x 10<sup>3</sup> km<sup>3</sup> in the southern hemisphere winter season (not shown). During hemispheric summer seasons, the sea-ice volume drops to 5 x 10<sup>3</sup> km<sup>3</sup> in the Arctic (PIOMAS ~15 x 10<sup>3</sup> km<sup>3</sup> September values during the 1980s) (Schweiger et al., 2011) and 0.5 x 10<sup>3</sup> km<sup>3</sup> in the Antarctic region. The 160/40 configuration generally produces much more sea ice compared to 80/20 (Fig. 2d red curves), which can also be inferred from the surface temperatures in high latitudes (Fig. 3a). In fact, since the PIOMAS reanalysis depicts the current state of the climate, the preindustrial sea-ice thickness is expected to be larger. The state of the ocean circulation of the two configurations is described by the overturning circulations in the Atlantic and Indo-Pacific regions (Fig. 5) and transport through various ocean passages that are important for various climate sub-systems (Table 3). For the last 500 years of simulation, the overturning circulations in the Atlantic at 26° N and 1000 m depth show values between 14-17 Sv for 80/20 and 16-19 Sv for 160/40, which is comparable to the RAPID array (~17 +/-4 Sv) (Frajka-Williams et al., 2019). The two configurations show a mono-cell structure with a northward transport of water masses in upper and mid-levels and southward transport in deeper levels. In the Pacific, the surface values indicate the subtropical cells at the northern and southern hemisphere. At deeper levels a basin-wide mid-depth outflow occurs in both configurations. The transports through the passages in both configurations are mostly simulated within the observational uncertainty found in the literature (see Table 3 for values and references). The transport through Bering Strait is a key element of the Arctic freshwater budget, and the values are close to the estimates by Woodgate et al. (2006) and Woodgate et al. (2012). The exchange of water masses between the Atlantic Ocean and the Nordic Seas plays a vital role in driving the global overturning circulation. The simulated transport rates are consistent with the circulation pattern described by Hansen et al. (2008). Similarly, the Indonesian Throughflow is a key component of the warm-water branch of the global conveyor belt. Although the

338339

simulated transport in this region is slightly underestimated compared to the values reported by Gordon et al. (2010), it still aligns reasonably well with observational estimates. These transports are similar to what is found in MPI-ESM (cf Table 5 in Müller et al., 2018) and ICON-ESM (cf Table 4 in Jungclaus et al., 2022). The Drake Passage transport is notably underestimated in 80/20, both when compared to the traditional estimate of around 135 Sv (Cunningham et al., 2003; Nowlin Jr. and Klinck, 1986) and to the more recent compilation by Donohue et al. (2016).

Figure 3: Near-surface temperature bias for a 30-year time slice of CTRL for (a) 160/40 and (b) 80/20. As reference ERA5 for the period 1979-2008 is used. Units are [°C].

Figure 4: Average sea-ice thickness for 80/20 for (a, b) the northern and (c, d) southern hemisphere for (a, c) winter (December, January, February - DJF) and (b, d) summer (June, July, August- JJA). The same 30-year time window of CTRL as in Figure 3 is used. Units are in [m].

Figure 5: The overturning circulation in the Atlantic (a,b) and Indio-Pacific (c,d) for 160/40 (a,c) and 80/20 (b,d). For both, the same 30-year time window of CTRL as Figure 3 is used. Units are in Sverdrup [ $10^9$  kg s<sup>-1</sup>].

The state of the ocean circulation in the North Atlantic is closely related to the deep-water mixing in the Labrador Sea and Irminger Sea, and at higher latitudes in the Norwegian and Greenland Seas. The deep convection of the Labrador Sea and Irminger Sea can drive the deep-water formation, and is suggested to impact on the AMOC. The mixing in the Norwegian and Greenland Seas contribute to the Arctic overflows and Atlantic bottom water. The mixed-layer depth in March is used here as a proxy for deep-water mixing (Fig. 6). It shows that the 80/20 configuration provides deep mixed layers in the Labrador Sea with maximum values of up to 2500 m. In the Irminger Sea, the mixed-layer depth reaches values of up to 1000 m. The maximum of the deep mixed layers in the 160/40 configuration is shifted to the Irminger Sea and reaches values of about 2500 m. The shift of the maximum values of the mixed-layer depth is closely related to the production of sea ice, which is larger in this configuration compared to the 80/20 configuration (see Fig 1d). The values of mixed-layer depths are generally higher compared to recent climate estimates for which maximum values of ~1000 m in the Labrador Sea and Irminger Sea are suggested (e.g., Königk et al., 2021). Finally, the mixed-layer depths in the Norwegian Sea are similar in both configurations and reach values of up to 2000 m.

Figure 6: The mixed-layer depth in March of CTRL for (a) 160/40 and (b) 80/20. The MLD criterion ("mlotst" model diagnostic) is the difference threshold of 0.03 kg m<sup>-3</sup> in potential density increase from the surface ocean. Units are in meters [m].

Ocean biogeochemical parameters for the 80/20 configuration are shown in Fig. 7. Average phosphate concentrations, total alkalinity, and dissolved inorganic carbon (DIC) at the surface are compared to the Global Ocean Data Analysis Project (GLODAP) version 2 database (Olsen et al., 2016). The spatial patterns of biogeochemistry fields are captured, with bias patterns similar to other Earth System models and previous ICON-ESM simulations (Jungclaus et al., 2022). Surface phosphate concentration is underestimated in the eastern equatorial Pacific and Southern Ocean, and overestimated along the southern Chilean coast. The bias in surface alkalinity and DIC is relatively small in most regions, with higher biases observed in coastal regions due to under-representation of coastal carbon dynamics (Mathis et al., 2022). The global pattern of surface alkalinity bias follows the bias in sea-surface salinity, with negative salinity bias leading to negative alkalinity bias. Since the model is forced with constant preindustrial atmosphere CO<sub>2</sub>, the surface DIC in the model adjusts to the surface alkalinity. Therefore, the bias in surface alkalinity is compensated by the bias in surface DIC, maintaining a correct ocean *p*CO<sub>2</sub> field. The simulated global flux of CO<sub>2</sub> into the ocean is approximately 0.1 PgC yr<sup>-1</sup>, close to the equilibrium levels at pre-industrial conditions.

Figure 7: Simulated phosphate concentrations (upper), surface total alkalinity (middle) and DIC (lower) for 80/20 climatology (left) and corresponding difference to reference data from the Global Ocean Data Analysis Project version 2 database (right). The GLODAP phosphate and alkalinity values are climatological means and the DIC is from pre-industrial estimates. The analysis is based on a 30-year time window of the CTRL experiment.

The performance of the land carbon model is illustrated by the gross primary productivity (GPP) for CTRL simulations in the 160/40 and 80/20 configurations (Fig. 8). The spatial GPP patterns in both configurations look very similar, with tropical productivity being much higher

than extra-tropical productivity. The patterns reflect the simulated biases in tropical precipitation (e.g., over eastern and central South America), but are otherwise very similar to the pattern simulated with MPI-ESM in CMIP6. The total annual productivity fluxes are 114.5  $\pm$  1.8 PgCyr<sup>-1</sup> and 112.9  $\pm$  1.6 PgCyr<sup>-1</sup> in the 160/40 configuration and the 80/20 configuration, respectively. Both model configurations are well within the CMIP6 model range for the preindustrial period and close to the pre-industrial GPP estimate of 113 PgCyr<sup>-1</sup> (Canadell et al., 2021).

Figure 8: 30-years mean of yearly accumulated gross primary productivity (GPP) for CTRL in the (a) 160/40 and (b) 80/20 configurations.

**Table 3:** Simulated and observed net volume transports across sections (positive means northward). Units are in [Sv].

| Ocean Passage                                                        | 160/40 | 80/20 | Observations          |
|----------------------------------------------------------------------|--------|-------|-----------------------|
| Bering Strait<br>(Woodgate et al., 2006;<br>Woodgate et al., 2012)   | 1.0    | 1.1   | 0.7–1.1               |
| Fram Strait<br>(Fieg et al., 2010)                                   | -1.5   | -2.0  | -1.75 ± 5.01          |
| Danmark Strait<br>(Hansen et al., 2008;<br>Jochumsen et al., 2012)   | -5.0   | -5.2  | -4.8;<br>-3.4 ± 1.4   |
| Iceland-Scotland<br>(Hansen et al., 2008; Rossby<br>and Flagg, 2012) | 4.9    | 5.1   | $4.8;$ $4.6 \pm 0.25$ |
| Indonesian Throughflow<br>(Gordon et al., 2010)                      | 12.4   | 12    | 11.6–15.7             |

152.1

111

 $134.0 \pm 14.0;$  $173.3 \pm 10.7$ 

#### 3.3 Transient Climate - 1850 to present

To recreate the climate of the historical period from 1850 to present, we employed external forcings from CMIP6, as the CMIP7 input data were not yet available at the time of these experiments. Specifically, we included yearly anthropogenic land cover changes, volcanic aerosol, and anthropogenic aerosol, which were added to the baseline aerosol concentrations of the pre-industrial period. Additionally, monthly ozone data and annual greenhouse gas concentrations were incorporated to reflect the evolving atmospheric composition over time. All experiments were conducted using the parameters derived from the CTRL experiments (see Section 3.2). For each of the configurations, a small ensemble of three members was generated. Each ensemble member was initialized from the corresponding CTRL experiment. The members differ only in their starting points, which were selected from various time points with the distance of 50 (160/80) and 25 years (80/20) apart in the CTRL period. Figure 9 shows the temporal evolution of GMT and global mean total precipitation. The development of GMT is close to observations from the 1960s onwards, and in the 2010s is about ~1.2 °C above 1850-1900. The increase is in the range of observed warming of 0.9-1.2 °C (Gulev et al., 2021). The global mean total precipitation shows a substantial positive bias in both configurations compared to GPCP and ERA5, and is on the upper end of all CMIP6 models. The global distribution of the bias reveals a strong double-ITCZ in the tropical Pacific with values up to 6 mm/day within the southern hemispheric branch, and particularly high values in the tropical Atlantic. Over the tropical continental regions strong dry bias occurs, such as in the Amazon region and over Indonesia (Fig. 9d and f). The precipitation bias in the tropical Pacific imposes a limitation for the global climate because it covers a large region of the globe in a rain-dominated area. Although the causes of the double-ITCZ are currently unclear, some models have modified the clouds microphysics, vertical entrainment rates, convection schemes or the atmospheric energy balance to reduce this feature (e.g., Ma et al., 2023; Ren and Zhou, 2024); however, no generalized modification can be applied to all models. In addition, we show the vertical temperature bias compared to ERA5 for the two

configurations (Fig. 10). The bias structures are characterized by cold biases of the tropical

atmosphere above the boundary layers, cold biases at tropopause levels, and warm biases at the surface in the high latitudes. The tropical cold bias reaches up to -1 °C in the 160/40 configuration accompanied with upper-level positive biases in the sub-tropics. The cold bias in 80/20 is increased up to -2 °C and reaches the sub-tropical regions. The positive surface bias is relatively large over the Southern Hemisphere with values up to 5 °C in both configurations and are in line with surface temperature distribution in Figure 3.

Figure 9 Evolution of (a) the global mean near-surface temperature (Kelvin) and (b) the global mean total precipitation (mm day<sup>-1</sup>) from the three historical ensembles HIST for the 160/40 (orange) and the 80/20 configuration (blue). The evolutions are compared with CMIP6 models (grey) and respective observations/reanalyzes (black). Geographical distribution of absolute values of (c, e) total precipitation and (d,

f) precipitation bias with respect to ERA5 for one member of the (c, d) 160/40 and (e, f) 80/20 configuration, averaged for the period 1979-2008 both in (mm/day). Details on reference data sets are given in Table 4.

Figure 10: (a) Annual mean zonal mean temperature in the troposphere for the period 1979-2008 for ERA5 and biases for HIST for one member of the (b) 160/40 and (c) 80/20 resolutions. Units are [°C].

A summary of the model performance is given in Fig. 11, which compares several key dynamical and thermodynamical variables with the CMIP6 model ensemble. Smaller root mean squared errors (RMSE) are found for many dynamical and thermodynamical quantities by increasing the resolution from the 160/40 to the 80/20 configuration. A similar impact of resolution is found for previous model versions, such as for MPI-ESM (Müller et al., 2018). Exceptions to the reduction of RMSE with resolution are variables describing the cloud properties and liquid water path, which underlines a systematic bias in the configurations with respect to the long-term mean hydrosphere. The 80/20 ensemble exhibits a relatively strong performance among the CMIP6 models for dynamical variables, such as zonal wind and temperatures in the mid- and upper troposphere.

Figure 11: The performance matrix for the 160/40 and 80/20 configurations (rightmost columns) and CMIP6 models (left columns) for key dynamical and thermodynamical variables. Shown are normalized relative space time root mean square errors (RMSEs) of the climatological seasonal cycle with respect to reference observational data sets. The normalization is done relative to the ensemble median of all models, with positive values (red) denoting a higher RMSE and thus worse performance, while negative values (blue) denote a lower RMSE than the ensemble median and thus a better performance. The considered time period is 2000-2014 for the models, for the observational reference data the time period had to be adjusted to the available time frame (see Table 4 for details). Boxes with a diagonal split indicate that two different reference data sets are used, with the first mentioned reference in the top left corner. The variables shown are the absorbed solar radiation (asr; reference: CERES-EBAF), ice water path (clivi; references: ESACCI-CLOUD, MODIS), total cloud cover (clt; references: ESACCI-CLOUD, PATMOS-x), condensed water path (clwvi; references: MODIS, ESACCI-CLOUD), specific humidity at 400hPa (hus400; reference: ERA5), liquid water path (lwp; references: ESACCI-CLOUD, MODIS), total precipitation (pr; references: GPCP-SG, ERA5), water vapor path (prw; reference: ESACCI-WATERVAPOUR), TOA outgoing longwave radiation (rlut; reference: CERES-EBAF), TOA outgoing shortwave radiation (rsut; reference: CERES-EBAF), temperature at 200 hPa (ta200; reference: ERA5) and 850 hPa (ta850; reference: ERA5), surface temperature (tas; references: HadCRUT5, ERA5), zonal wind stress (tauu; reference: ERA5), and zonal wind at 200 hPa (ua200; reference: ERA5) and 850hPa (ua850; reference: ERA5).

**Table 4:** Observational reference data sets used in Fig. 11.

| Reference data sets    | Туре              | Variables                                                                                                                                         | Time range<br>used in<br>Figure 7 | Reference              |
|------------------------|-------------------|---------------------------------------------------------------------------------------------------------------------------------------------------|-----------------------------------|------------------------|
| CERES-EBAF Ed4.2       | Satellite         | Absorbed solar radiation (asr) TOA outgoing longwave radiation (rlut) TOA outgoing shortwave radiation (rsut)                                     | 2001-2014                         | Loeb et al. (2018)     |
| ERA5                   | Reanalysis        | Specific humidity (hus) Total precipitation (pr) Air temperature (ta) Near-surface air temperature (tas) Zonal wind stress (tauu) Zonal wind (ua) | 2000-2014                         | Hersbach et al. (2020) |
| ESACCI-CLOUD           | Satellite         | Ice water path (clivi) Condensed water path (clwvi) Total cloud cover (clt) Liquid water path (lwp)                                               | 2000-2014                         | Stengel et al. (2020)  |
| ESACCI-<br>WATERVAPOUR | Satellite         | Water vapor path (prw)                                                                                                                            | 2003-2014                         | Schröder et al. (2023) |
| GPCP-SG v2.3           | Satellite - gauge | Precipitation (pr)                                                                                                                                | 2000-2014                         | Adler et al. (2017)    |

| HadCRUT5 v5.0.1.0 (analysis) | Ground     | Near-surface air temperature (tas)                                          | 2000-2014 | Morice et al. (2021)    |
|------------------------------|------------|-----------------------------------------------------------------------------|-----------|-------------------------|
| MERRA2                       | Reanalysis | Near-surface air temperature (tas)                                          | Not used  | Gelaro et al. (2017)    |
| MODIS                        | Satellite  | Ice water path (clivi) Condensed water path (clwvi) Liquid water path (lwp) | 2003-2014 | Platnick et al. (2003)  |
| PATMOS-x                     | Satellite  | Total cloud cover (clt)                                                     | 2000-2014 | Heidinger et al. (2014) |

### **3.4 Climate Sensitivity**

Climate sensitivity describes the response of the climate system to radiative forcing and is a critical parameter that determines the future evolution of climate. Two metrics are commonly used: the transient climate response (TCR) and the equilibrium climate sensitivity (ECS).

TCR is determined from the 1%CO<sub>2</sub> experiment as the global mean surface air temperature increases (relative to the CTRL experiment) around the time of doubling CO<sub>2</sub>. Following Meehl et al. (2020) and Jungclaus et al. (2022), a 20-year average is taken around the doubling of CO<sub>2</sub> in order to reduce the potential influence of internal variability. The TCR is 1.7 K for the 160/40 configuration and 1.6 K for the 80/20 configuration (Fig. 12a and b). The assessment of climate sensitivity in CMIP6 models shows a best estimate of TCR=1.8 K with a very likely range of 1.2 to 2.4 K.

ECS is approximated with the so-called "effective climate sensitivity" (Gregory, 2004) using an idealized experiment where the atmospheric CO<sub>2</sub> concentration is abruptly quadrupled (4xCO<sub>2</sub>). For this, a linear regression is applied between the global mean surface air temperature change (relative to the CTRL experiment) and the net downward radiative flux at the top-of-atmosphere over 150 years of the simulation (see Fig. 12c and d). The extrapolation of the regression line to zero net radiation gives the temperature response with quadruple increase in CO<sub>2</sub>, which is then divided by two to get an estimate for the ECS. This results in an ECS of 2.47 K for both 160/40 configurations. The assessment of climate sensitivity in CMIP6 models shows a best estimate of ECS=3 K with a very likely range of 2 to 5 K (Forster et al., 2021). The climate sensitivity of ICON XPP falls within these CMIP6 ranges, tending towards the lower end of the spectrum.

Figure 12: Estimating climate sensitivity. The Transient Climate Response (TCR) is estimated from the global mean surface air temperature anomaly at the time of CO<sub>2</sub> doubling (at year 70) in the 1%CO<sub>2</sub> experiment for (a) 160/40 and (b) 80/20. The Equilibrium Climate Sensitivity (ECS) as diagnosed from the scatterplot between TOA net radiance and global mean surface temperature anomaly, including a linear regression for (c) 160/40 and (d) 80/20. ECS is estimated from 150 years of the 4xCO<sub>2</sub> experiments (black line), but since the assumption of linear feedback is only an approximation, the regression lines and the estimated ECS values for the first 20 years (blue line) and the last 130 years (orange line) are shown for completeness.

# 4. Key dynamical processes in the tropics, extra-tropics and stratosphere

ICON XPP is intended to be the successor of MPI-ESM for climate prediction research and operational forecasts. A principal foundation of climate predictions is based on the reliable description of the major modes of climate variability and their associated background mean state. Examples of such modes of variability are the Madden-Julian Oscillation (MJO), ENSO,

and the Quasi-Biennial Oscillation (QBO) in the tropics, or the NAO and its relation to the extra-tropical jet position in the extra-tropics. While designing the model configurations, we therefore put special emphasis on monitoring certain aspects of the mean climate which are directly related to the major modes.

### 4.1 Tropics

In contrast to the mid-latitudes, the release of latent heat is the main source of energy in the tropical atmospheres. This occurs in conjunction with convective cloud systems embedded in large-scale circulations. The diabatic heating associated with tropical precipitation not only leads to a localized response in the atmospheric circulation, but can also cause a remote response through the excitation of equatorial waves.

#### 4.1.1 Tropical Waves and Madden-Julian Oscillation

- Equatorially trapped waves are a fundamental property of tropical dynamics and appear as solutions of the shallow water equations which are either symmetric or asymmetric about the equator. Among others, the observed disturbances in the clouds can be associated with equatorial trapped waves (Wheeler and Kiladis, 1999). By creating the wavenumber-frequency spectrum of the outgoing longwave radiation (OLR), modes of tropical variability can be analyzed in more detail (Wheeler and Kiladis, 1999). We use the OLR as it is generally assumed that is a reasonable proxy for deep tropical convection and precipitation.
- The principal nature of the tropical spectrum is red in both zonal wavenumber and frequency, with highest power at the lowest frequency and lowest zonal wavenumber. Thus, an estimated background spectrum is removed prior to the analysis of tropical waves. Typically, the peaks then follow the dispersion curves of equatorial trapped waves. Most of the preferred modes of variability are observed in the symmetric component, such as the MJO (eastward zonal wavenumber 1-5, frequencies of about 

Figure 13: Wavenumber-frequency OLR spectra for the symmetric components (top) and asymmetric components (bottom) averaged between 15° S and 15° N for (a,d) ERA5, (b,e) 160/40 and (c,f) 80/20. Solid lines represent the dispersion curves of the odd (top) and even (bottom) meridional mode-numbered equatorial waves for the three equivalent depths of h = 12, 25, and 50 m [as in Wheeler and Kiladis, 1999]. For ICON XPP, high frequency output of a 10-year period (2000-2010) of one realization is used for both configurations.

#### 4.1.2 El Niño/Southern Oscillation (ENSO)

The El Niño/Southern Oscillation (ENSO) is one of the key processes for climate predictions on seasonal to annual time scales, and is routinely predicted in numerous operational forecast systems. However, ENSO is determined by the complex interplay of the mean climate state in the tropical Pacific, the internal ENSO dynamics (Guilyardi et al., 2020), and also by global remote influences, for example the Atlantic and Indian Oceans (Cai et al., 2019). In many

forecast systems and their underlying Earth system models, the mean state and trends of the tropical Pacific - and thus the ENSO dynamics - are only inadequately represented (Guilyardi et al., 2020). CMIP-like models show long-term mean errors ("cold tongue bias") and strongly underestimated ENSO feedbacks. The cold tongue bias refers to the excessive cooling along the equatorial Pacific, a common systematic error in climate models (Li and Xie, 2014). The MPI-ESM, for example, clearly has weak Bjerknes feedbacks and atmospheric damping in conjunction with a strong tropical Pacific cold bias (Bayr et al., 2018). This has an impact on the simulated development of an ENSO event. A balanced interplay between the mean state and the ENSO dynamics in the tropics can therefore be assumed as a basic prerequisite for successful ENSO predictions. We investigate ENSO during the tuning process with a particular focus not only on isolated ENSO performance (e.g., amplitude, seasonality), but also consider the ENSO dynamics (feedbacks) and the ENSO relation to the mean state bias. We apply the ENSO metric package developed by CLIVAR, which is designed to evaluate the model with respect to the basic state, ENSO performance and their feedbacks, as well as the ENSO teleconnections (Planton et al., 2021). A regression of SST anomalies to the Nino3.4 index for both configurations clearly exhibit an ENSO pattern in the tropical Pacific for both configurations (Fig. 14). The strongest anomalies are found in the central-to eastern Pacific similar to the reference. However, as in many coupled models, the ENSO activity in ICON XPP exhibits a stronger westward extension of the SST anomalies than observed (Capotondi et al., 2020). Figure 15 gives more details of ENSO for the two configurations. Fig. 15a shows a general summary of several metrics from the CLIVAR ENSO package and illustrates ENSO-related mean states, performance, feedbacks and teleconnections in ICON XPP relative to the CMIP6 models. The values within the box indicate that ENSO in ICON XPP is within 90% confidence intervals of the CMIP6 model ensemble. Positive values that are outside the box show that the experiments have a significantly weaker performance than the CMIP6 models. Clearly, for the

performance and feedbacks metrics ENSO in ICON XPP is within the range of the CMIP6 models. The ENSO-related mean state and teleconnection summary, however, indicates a

larger bias compared to the CMIP6 ensemble. A general improvement is found for all metrics

for higher resolution experiments (80/20) compared to the 160/40 runs.

We further examine ENSO by looking into the individual metrics. The mean SST illustrates that the model configurations are about 1.5-2 °C colder than the reference, mainly in the

western and central Pacific, associated with the cold-tongue bias (Fig. 15b). The west-east SST gradient is about 4 °C and the SST slope is close to what is shown in the TropFlux reference. In the western Pacific edge (150° E-160° E), the SST gradients are relatively steep in both configurations. In the eastern Pacific edge (240° E-270° E), the SST gradient reverses in both configurations. The ENSO-related zonal wind stress substantially improves in the higherresolved configuration compared to the 160/40 resolution (Fig 15c). In 80/20 the magnitudes are much closer to the reference, and the minimum is shifted eastward closer to what is observed. In addition, we show the zonal mean total precipitation for the Pacific (Fig. 15d). The distributions clearly reveal a double-ITCZ in both configurations, with a strong deviation from observations shown in the Southern Pacific. The bias is relatively large in both configurations with values up to 4-5 mm day<sup>-1</sup>. The double ITCZ bias is found in many coupled models, and is linked with their ENSO characteristics, such as the ENSO seasonal phaselocking (Liao et al., 2023). The ENSO characteristics of the two configurations are shown in Figure 15e-g. The ENSO amplitude - defined as the standard deviation of SST anomalies - across the equatorial Pacific shows weaker values in the eastern part and stronger values in the western part (Fig. 15e). The amplitude of the Nino3.4 index appears a bit weak and is about 2/3 of the observational amplitude. During the peak season of ENSO the Nino3.4 index is about 0.7 and 0.8 °C in 160/40 and 80/20 compared to 1.2 °C in TropFlux (Fig. 15f). In addition, the ENSO skewness shows larger (smaller) values in the western (eastern) Pacific and indicates a western shift of the peak ENSO (Fig. 15g). Finally, the ENSO feedbacks are shown in Fig. 15h-j. A positive wind stress-SST relationship explains an anomalous zonal wind with the SST propagation along the equatorial Pacific. For example, during El Niño, a stronger wind stress anomaly (weaker trade winds) is associated with eastward propagation of SST anomalies. This relationship is captured in both configurations, but with less amplitude and the maximum regression coefficients appear shifted eastward compared to observations (Fig. 15h). The wind stress is furthermore related to thermocline depth, meaning that for example during El Niño there is a shallowing (deepening) of the thermocline depth in the western (central-to-eastern) Pacific (Fig. 15i, here the thermocline depth is illustrated by the sea surface height. In both configurations, the negative wind stress-SSH relationship in the western Pacific is absent, while positive regression coefficients are found in the central-to-eastern Pacific. In the central-to-eastern Pacific, the

80/20 configuration shows regression coefficients closer to observation. Finally, the negative SST-heat flux relationship illustrates the atmospheric damping effect, i.e. in case of El Niño, a warm SST anomaly results in a stronger updraft and cloud cover increase which in turn reduces the net incoming radiation at the surface (Fig. 15j). In ICON XPP, this feedback is strongly underestimated which reflects a systematic bias in the heat fluxes, in particular in the centralto-western Pacific. This is a common bias found in many CMIP models, in which a weak atmospheric heat flux damping compensates the weak Bjerknes feedback (Bayr et al., 2018). The weak SST-heat flux relationship in ICON XPP is dominated by the shortwave radiation fluxes (not shown), similar to what is found in other models (Bayr et al., 2018). In summary, ICON XPP generates an ENSO with typical characteristics and dynamics known from observations. However, ICON XPP performs weaker amplitudes and feedbacks compared to observations with the current parameter setting, but an improvement is found for 80/20 compared to 160/40. We also find structural biases similar to the long-standing errors of many coupled models. Here, the overall performance with respect to the CMIP6 models reveal pronounced biases in both configurations, closely associated with the precipitation bias. However, in other key diagnostics – performance and feedbacks - ENSO in ICON XPP is within the range of the CMIP6 models. It is worth noting that in some aspects both configurations share similar features. An example is the precipitation bias which clearly indicates a pronounced double-ITCZ, or the weak ENSO amplitudes in both configurations. This points towards systematic errors covered in both configurations. Thus, the much faster and cheaper configuration can be used to more easily explore the space of hyperparameters to identify potential tuning improvements for the ENSO representation. First attempts point towards the role of cloud properties and microphysics in modulating the surface radiation budget that affect the atmospheric damping and the SST- wind stress feedbacks.

Figure 14: Regression between the Nino3.4 index and SST anomalies (SSTA) for December for one member of (a) 160/40, (b) 80/20, and (c) and TropFlux. The Nino3.4 index is defined as the area-averaged SST anomaly over 5° N to 5° S and 170° W to 120° W. Also shown are the differences between (d) 160/40 and TropFlux, and (e) 80/20 and TropFlux. Units are in [°C/°C]. The regression is calculated with the CLIVAR ENSO metric package (see Planton et al. (2021) for details). As reference in (c-e) TropFlux is used (Praveen Kumar et al., 2012). TropFlux consists of daily and monthly fluxes, SST and wind stress for the tropical region for 30° S to 30° N, and combines ERA-Interim and ISCCP corrected using Global Tropical Moored Buoy Array data from 1979 to present.

Figure 15: Description of ENSO. (a) An overall summary of different categories of the ENSO metrics (climatology, characteristics, feedbacks and teleconnections) for (red) the 160/40 and (blue) 80/20 ensemble members together with the CMIP6 models. See Planton et al. (2021) for all metrics and their definitions. Further shown are specific metrics for (b-e) ENSO-related climatology, (e-g) the ENSO characteristics, and (h-j) the ENSO feedbacks for all ensemble members of (red) the 160/40 and (blue) 80/20 configurations, and (black) an observational reference. The mean states are illustrated by (b) SST averaged for 5° N to 5° S, (c) zonal wind stress averaged for 5° N to 5° S, and (d) precipitation averaged for 150° W to 90° W. The ENSO characteristics are described by (e) the zonal structure of the standard deviation of the Nino3.4 SST anomalies (SSTA) averaged for 5° N to 5° S, (f) the standard deviation of SSTA as a function of calendar months, (g) the skewness of SSTA in the equatorial Pacific averaged for 5° N to 5° S. ENSO is further analyzed by the Bjerknes feedbacks, here shown

by (h) the regression of zonal wind-stress anomalies (meridional 5° S to 5° N average) onto SSTA in the eastern equatorial Pacific (Niño3 region averaged), and (i) the regression of sea-surface height (SSH) anomalies (meridional 5° S to 5° N average) on to wind-stress anomalies (Niño3 region averaged). The atmospheric damping is illustrated by (j) the regression of the total atmospheric surface heat flux anomalies on SSTA, both 5° N to 5° S averaged. The plots are calculated based on the CLIVAR ENSO metric package (Planton et al., 2021). As references this package uses GPCPv2.3 for precipitation, AVISO for SSH, and TropFlux otherwise (Praveen Kumar et al., 2012).

#### 4.2 Extra-tropics - Zonal mean zonal wind and jets

The extra-tropical jets provide a substantial guideline for synoptic-scale disturbances. Among others, the extra-tropical storm paths are aligned to the position and magnitude of the seasonally and yearly varying jet positions and impact weather and climate further downstream. In addition, the time-averaged tropospheric jets act as a wave-guide for Rossby-like traveling waves propagating from the tropical regions to the extra-tropics, and thereby have a control on the mid-latitude dynamics (Branstator, 2002). In the extra-tropics, the zonal and meridional jet variation are closely linked to the major modes of climate variability, i.e. the NAO (Woollings et al., 2015), and its predictability (i.e., Strommen et al., 2023). The NAO constitutes a principal driver of the North Atlantic and European climate, in various prediction systems and underlying coupled models (Doblas-Reyes et al., 2003), and seasonal and decadal prediction skill of the NAO is established (Athanasiadis et al., 2020; Dobrynin et al., 2018; Müller et al., 2005; Scaife et al., 2014; Smith et al., 2020).

However, climate models still provide biases in the representation of the zonal wind, and associated jets and storm tracks. For example, CMIP6 models are generally able to reproduce

However, climate models still provide biases in the representation of the zonal wind, and associated jets and storm tracks. For example, CMIP6 models are generally able to reproduce storm tracks, however, they appear too zonal over the Pacific and Atlantic (Priestley et al., 2020). Over the southern hemisphere, climate models tend to shift jet positions and storm tracks too far equatorward. There is a general improvement of the biases from CMIP5 to CMIP6, which arises from the tendency of using higher model resolutions, but their bias structures still persist (Priestley et al., 2023). In MPI-ESM used for CMIP6, the mean zonal wind and storm track biases are reduced by doubling the atmospheric resolutions. The bias reduction is mainly induced by an improved wave-activity flux and eddy-driven effects on the mean zonal wind, particularly at the exit of the Northern hemisphere jet (Müller et al., 2018). However, a relatively strong zonal wind bias persists in the higher-resolved model version. In this respect,

the underestimation of eddy-driven effects on the mean zonal wind is found in many climate

models (i.e., Smith et al., 2022).

In ICON XPP, the zonal mean zonal wind biases in the extra-tropics appear smaller compared to its predecessors ICON-ESM (Jungclaus et al., 2022) and MPI-ESM (Müller et al., 2018). In the 80/20 configuration, a zonal mean zonal wind bias of 1-2 m s<sup>-1</sup> is found at the northern hemisphere jet position (Fig. 16), and of 2-4 m s<sup>-1</sup> in the low resolution (160/40). For comparison in MPI-ESM, zonal mean zonal wind biases are about twice as large and amount to 2-4 m s<sup>-1</sup> and >4 m s<sup>-1</sup> for similar resolutions compared to ICON XPP (cf Fig. 9 in Müller et al., 2018). For ICON-ESM, a bias of up to 10 m s<sup>-1</sup> is found in their 160/40 configuration (cf Fig. 12 in Jungclaus et al., 2022). In the tropics, there are alternating significant positive and negative wind biases varying with height. The biases are smaller compared to ICON-ESM but

of similar magnitude compared to MPI-ESM.

To understand the reasons for the relatively small biases of the northern hemisphere zonal

To understand the reasons for the relatively small biases of the northern hemisphere zonal winds, we further examine the eddy-mediated effects on the jets. Figure 17 shows the mean zonal wind at a level where the jet maximum occurs and corresponding divergence of 2-6 day bandpass-filtered eddy-momentum fluxes. The divergence is calculated based on the horizontal components of the E-vector averaged over 200-300 hPa (Hoskins et al., 1983). The net effect of the divergence is a westerly acceleration of the mean flow, whereas a convergence is associated with increased easterlies. ERA5 reveals divergence of the E-vector downstream of the maximum zonal wind, which indicates that momentum fluxes are able to force the jets towards the north-eastward direction. In ICON XPP, eddy-momentum fluxes are found similar to ERA5 and the jet is forced towards a north-eastward direction. That is different to precursors of ICON XPP, where momentum fluxes and respective jets appear more zonally oriented. The magnitudes of the divergence of the momentum fluxes in 160/40 are higher than in ERA5, but fits very well in 80/20. This diagnostic underlines the good performance of the synoptic properties in ICON XPP for the mean state of the jet. It can be expected that this has a positive impact on storm-track pathways and associated impact on downstream regional climate.

Figure 16: (a) Annual man zonal mean zonal wind in the troposphere for the period 1979-2008 for ERA5 and biases for the (b) 160/40 and (c) 80/20 resolutions. Units are [m s<sup>-1</sup>].

Figure 17: The effect of transient eddies on the mean state. Shown is the divergence of the E-Vector (shading) and the mean zonal wind (contours) for winter means (DJF) in (a) ERA5 for the period 2000-2010, and 10-year averages for the ICON XPP (b) 160/40 and (c) 80/20 resolutions. The E-Vector is calculated by  $\nabla(u^2 + v^2, -uv)$ , where u and v are 2-6 day band-pass filtered zonal and meridional wind anomalies. The E-vector and mean zonal wind are averaged between 200-300 hPa. Positive values of the divergence indicate a transfer of momentum to the mean state.

### 4.3 Stratosphere - Quasi-Biennial Oscillation

Developments in recent decades have shown that seasonal and long-range climate predictions

benefit from resolving the stratosphere at depth, as the variability of the stratosphere is not only

affected by the lower atmosphere and surface climate, but also by intrinsic interactions (e.g.,

Domeisen et al., 2020; Manzini et al., 2014; Scaife et al., 2022). The quasi-biennial oscillation

(QBO) is a key process in this respect.

The QBO is an important component of the Earth's climate, controlling equatorial zonal winds and temperature deviations from the global mean. Its teleconnections to surface climate occur in various pathways (Gray, 2018). In the tropics, a link between the tropical stratosphere and the MJO has been revealed as the phase of the QBO modulates the MJO (e.g., Martin et al., 2021). In addition, the QBO modulates the winter stratospheric polar vortex in the Northern Hemisphere, which has implications for the troposphere (Holton and Tan, 1982). Both the QBO and the variability of the stratospheric polar vortex are examples of predictability originating

in the stratosphere.

772

The observed QBO is characterized by descending alternating easterly and westerly jets in the tropical stratosphere and their downward propagation into the troposphere, as shown by the zonally averaged zonal wind (Fig. 18a). In the ICON XPP 160/40 configuration, the descending winds are weakly easterly with a high periodicity of roughly 12 months at 32 km (~ 10 hPa), compared to roughly 28 months in observations (Fig. 18b). For the higher resolution 80/20 a QBO is present and the period increases to 17 months, although the amplitudes still appear smaller than observations (Fig. 18c). A quasi-permanent easterly wind in the lower-to-middle stratosphere is prominent in both resolutions (Fig. 18b, c). In order to assess the QBO independently from the climatological state, the long-term mean is removed from the QBO time series (Fig. 18e, f). The zonal wind anomalies emphasize that ICON XPP is capable of developing spontaneous QBO phases and their downward propagation (Fig. 18e, f). However, in 160/40 with 90 vertical levels only, the QBO appears disruptive and the downward propagation is not well established (Fig. 18e). The long-term mean equatorial zonal mean wind in the model configurations further exhibit strong easterly winds at an altitude of about 20 km height. These easterly winds can act as a permanent wave filter for vertical wave propagation, resulting in a perturbed wave forcing above that height and hindering the QBO development in ICON XPP. The reason for the development of this easterly jet is unclear, but seems related to the horizontal resolution.

In atmosphere-only experiments (160 km, 130 levels), the frequency of the QBO phases has been examined in the past (Niemeier et al., 2023). In these experiments the QBO is well established and benefit from increasing the number of vertical levels. The lower vertical resolution of 90 levels is found too coarse to generate an internally generated QBO. Further, in the atmosphere-only experiments a much smaller time step was used, which seems to further improve the QBO (360 seconds in atmosphere-only experiments compared to 450 seconds in the coupled configurations). However, the reasons for such impact are yet not fully understood. In addition to the QBO, the atmosphere-only experiments reveal a well-represented stratospheric transport. As an example, the transport of the water vapor cloud after the Honga Tonga eruption is found very close to observations (Niemeier et al., 2023).

Figure 18: Zonal mean zonal wind averaged between  $5^{\circ}$  S and  $5^{\circ}$  N in (a) ERA5, (b) 160/40 with 90 vertical levels (L90) and (c) 80/20 with 130 vertical levels (L130) and (d), (e), (f) the corresponding deviation of the long-term mean (1979-2008). Here the period 1979-1989 is shown. Units are in [m s<sup>-1</sup>].

## 5. Discussion and Conclusion

ICON XPP is a newly developed Earth System model configuration based on the ICON modeling framework. It merges accomplishments from the recent operational numerical weather prediction model (ICON NWP) with well-established climate components for the ocean, land and ocean-biogeochemistry into a new Earth System model configuration. Here, we discussed two baseline configurations which serve as a starting point for accommodating ICON for Earth System predictions and projections, and future model development.

ICON XPP in the presented configurations reaches typical targets of a coupled climate simulation, such as a pre-industrial stable climate equilibrium with radiation balance and a target global mean temperature. Though the presented configurations share some long-standing biases typical for coupled models, such as warm biases in the coastal upwelling regions, the overall fidelity of ICON XPP fits in the CMIP6 ensemble. This is noteworthy since a major newly implemented component is the atmospheric model component ICON NWP, originally designed for numerical weather prediction, and which is tested here for the first time in a coupled Earth System configuration. Furthermore, the climate sensitivity, albeit weak, fits within the assessed range of the CMIP6 models, and creates confidence in ICON XPP projections. The simulated trends of the global temperature are close to observations and underline the model's suitability to simulate various climate scenarios.

The model configurations are able to capture the principal features of coupled circulations in the tropics. A prominent example is ENSO, which reveals typical characteristics and dynamics known from observations. We highlighted the use of a more sophisticated evaluation of ENSO, by not only looking at certain characteristics (amplitude, spectra, skewness, etc.), but also considering the ENSO dynamics (feedbacks) and its link to the mean bias. Although ENSO amplitudes and basic feedbacks appear weak, the overall fidelity of ENSO in ICON XPP fits within the CMIP6 models. Further examples of key processes in the tropics are the tropical waves and Madden-Julian Oscillation, which are captured quite well in both configurations. In addition, ICON XPP is capable of developing spontaneous QBO phases, which clearly benefits from the higher vertical resolution in the 80/20 configuration.

An outstanding result of the current evaluation is the state of the northern-hemisphere extratropical dynamics. Here, ICON XPP reveals a strong reduction of the tropospheric zonal mean zonal wind biases, and the location of the mean jets are placed close to what is found in observations. A closer examination of the synoptic-scale eddies reveals that ICON XPP is able to depict the shape and magnitude of the transfer of momentum onto the mean flow close to what is found in ERA5. The momentum transfer leads to a northeastward elongation of the mean jet in ICON XPP, whereas predecessor model generations reveal a strong zonal distribution. This could have consequences for the storm tracks and their downstream impacts, which are known to exhibit a biased southern pathway in the ICON XPP precursors. We hypothesize that this improvement is linked to the enhanced accuracy in resolving synoptic disturbances within the ICON NWP model.

However, the current ICON XPP configurations are characterized by some strong biases with global implications. These include a warm bias of up to 5° C in the Southern Ocean, associated with little sea ice. This is accompanied with a particularly deep ocean mixed layer at the Antarctic boundaries near the Weddell Sea and strong biases in the atmospheric net radiation and cloud covers (not shown). Errors of this magnitude inevitably lead to the need to adjust the model. In order to achieve the global mean temperature target, it was necessary to counterbalance the Southern Ocean warming by adjustment of cloud parameters, e.g. reducing the entrainment rate. Further, the Southern Ocean plays an important role in remote regions of the climate system. Recent studies reveal the global role of the observed Southern Ocean cooling trends and their teleconnections, such as to tropical regions and the southeast Pacific cooling (Kang et al., 2023). However, many climate models notoriously fail to capture the recent SST trend in the Southern Ocean. Also, all coupled model and climate prediction systems are not able to capture the Pacific cooling trend with consequences on forecasting the Pacific climate such as ENSO (e.g., L' Heureux et al., 2022). Therefore, an improvement of the Southern Ocean climate may be of great relevance for remote regional climate and their predictions. The tropical precipitation distribution reveals the long-standing double ITCZ, as found in many CMIP6-like models. In our configurations, however, the magnitudes are relatively large compared to the CMIP6 ensemble. Such a bias ultimately imposes an influence on regional and global climate. An example is ENSO, which has a strong relationship with the precipitation bias in the current ICON XPP configuration. Further, a strong dry bias in the Amazonian region is found in the current configurations. Such bias imposes an impact on the modeling of land vegetation and the global carbon cycle. The reasons for the tropical precipitation bias are yet unclear. However, since during the tuning process the precipitation distribution has not received much attention, we expect some improvements in subsequent versions of ICON XPP. ICON XPP forms the basis for future developments in the areas of climate predictions and projections. Some initiatives have already been established for this purpose. One project was initiated to support ICON XPP's preliminary research into climate predictions. Here, data assimilation methods and hindcasts are being tested with ICON XPP, as well as their possibilities for special applications. The aim is, among others, to use ICON XPP for operational climate predictions. Another initiative prepares ICON XPP as a national contribution to CMIP7. For this, ICON XPP will be more thoroughly tuned with respect to the

aforementioned biases. In addition, corresponding DECK experiments with CMIP7 forcing will be prepared and carried out. A basic requirement for both initiatives is that the model is able to calculate as many model years and ensemble members as possible, in as little real time as possible. The high runtime performance of the current configurations with throughput of ~80 simulated years per day (SYPD, 100 nodes) for 160/40 and ~45 SYPD (64 nodes) for 80/20 - run on a CPU-partition of the DKRZ HPC - meet this requirement.

In summary, ICON XPP is an Earth System model configuration, able to run long integrations and large-ensemble experiments, making it suitable for climate predictions and projections, and for climate research for which a large throughput is required.

889

## Acknowledgments

We would like to acknowledge the German Climate Computing Center (DKRZ) for providing the computing facilities used to run the DECK experiments and various test simulations. This research was supported by the German Ministry of Education and Research (BMBF) by the project Coming Decade (WM, DY - grant 01LP2327E) and CAP7 (MS - grant 01LP2401C). IP and JK acknowledge funding from the Hans Ertel Centre for Weather Research (Hans-Ertel-Zentrum für Wetterforschung; HErZ). HerZ is a German research network of universities, research institutions, and the German Meteorological Service (Deutscher Wetterdienst; DWD), and is funded by the Federal Ministry of Transport (Bundesministerium für Verkehr; BMV). JK received funding from the HErZ project OceanWeather (grant 4823DWDP1B). The DWD's "Innovation Program for Applied Research and Development - IAFE" funded AS through the project ICON-Seamless VH 4.7 project, and KCM through the project POINTS. Funding has been received by the German Research Foundation (DFG) under the Excellence Strategy -EXC 2037 'CLICCS - Climate, Climatic Change, and Society' (No 390683824, FC, TI). UN received support from the DFG Research Unit VolImpact (FOR2820, No 398006378) and from the project STATISTICS founded by the European Space Agency. DD was supported by the Australian Research Council (ARC) Centre of Excellence for Climate Extremes (Grant Number: CE170100023). CAK received funding from the ETH Postdoctoral Fellowship program- Model tuning work/GPU port testing was supported by the Swiss National Supercomputing Center (CSCS) under the project ID s1283. Several authors received funding from the European Union's Horizon Europe research and innovation program under grant agreement No 101003536 (project ESM2025, FC, TI, VB), No 101081460 (project ASPECT,

HP), and No 951288 (project Q-Arctic, VB, TS). Views and opinions expressed are, however, those of the author(s) only and do not necessarily reflect those of the European Union. Neither the European Union nor the granting authority can be held responsible for them. We thank Marco Giorgetta for reviewing the manuscript before submission.

## **Code and Data Availability Statement**

The run scripts and manual used to run ICON XPP for this study are available in the Open 926 Research Data Repository of the Max Planck Society (https://doi.org/10.17617/3.UUIIZ8) 927 (Müller et al., 2024). ICON is available to the community under a permissive open source 928 license (BSD-3C). Please follow the instructions on the ICON web-page (https://www.icon-929 model.org/). ERA5 data (Hersbach et al., 2020) was downloaded from the Copernicus Climate 930 Change Service (Hersbach et al., 2023a, 2023b). The results contain modified Copernicus 931 Climate Change Service information 2025. Neither the European Commission nor ECMWF is 932 responsible for any use that may be made of the Copernicus information or data it contains. 933 Figures 6a, 6b, 7, and 8 of this study have been created with the Earth System model Evaluation 934 Tool (ESMValTool) (Andela et al., 2024a; Righi et al., 2020) and its core dependency 935 ESMValCore (Andela et al., 2024b). ESMValTool has recently been extended to be able to process ICON XPP output without any model postprocessing (Schlund et al., 2023). CMIP6 936 937 model output required to reproduce the analyses of this paper is available through the Earth 938 System Grid Foundation (ESGF; https://esgf-metagrid.cloud.dkrz.de/search/cmip6-dkrz/, last 939 access: 19 February 2025). ESMValTool can automatically download these data if requested 940 (see 941 https://docs.esmvaltool.org/projects/ESMValCore/en/v2.11.1/quickstart/configure.html#esgf-942 configuration, last access: 19 February 2025). Observational/reanalysis datasets are not 943 distributed with ESMValTool that is restricted to the code as open source software, but 944 ESMValTool provides a collection of scripts with downloading and processing instructions to 945 recreate all observational/reanalysis datasets used for Figures 9a, 9b, and 11 (see 946 https://docs.esmvaltool.org/en/latest/input.html#observations, last access: 19 February 2025).

## **Authors Contribution**

- Conceptualization: WM, TvP, AS, SL. Writing original draft: WM, TvP, AS, SL, FC.
- Visualization: TvP, AS, SL, MS, DY, RB, TS. Model component development: VB, NB, FC,
- KF, VG, HH. SH, MH, TI, JJ, MK, PK, LK, CAK, JK, KC-M, UN, HP, IP, TR, TS, RW.
- Supervision: JM, RP, BF.
- Competing Interests
- The contact author has declared that none of the authors has any competing interests.

**Disclaimer**

**Review Statement**

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
