# Peer review of "The ICON-based Earth System Model for Climate"

_EGUsphere, 2025_

## Author Comment (AC1)

We sincerely thank the reviewer for the time he or she spent editing the manuscript. We would also like to thank them for the numerous comments, some of which highlighted passages that were certainly still unclear or incomplete. We believe this has significantly improved the paper.

Our answers are integrated into the review below.
* * *
Review #1

"This manuscript presents ICON XPP, the first version of a new climate model specifically tailored for climate prediction applications at two alternative model resolutions.

Both the model documentation and the evaluation of the two resolution configurations are comprehensive, spanning seasonal to decadal scales and covering important drivers of predictability such as the ocean and atmospheric circulations, the stratosphere and key modes of internal climate variability. The authors employ a wide range of diagnostics and benchmark against well-established datasets, demonstrating the model's strengths and identifying areas for improvement.

I find the paper to be of interest and suitable for publication in GMD, pending the authors' response to a few clarifications and minor comments listed below. "

- **The manuscript is generally clear but presents frequent typos and grammatical errors. I recommend a thorough proofreading to improve its readability.**

  Thanks a lot for pointing at this. We will have a thorough check, for example proof reading by a few co-authors. In relation to reviewer 2, we also will sharpen the abstract and restructure the introduction.

- **Sentence in lines 105-109: The final part of the sentence seems to be incomplete. Did you mean to say that MPI-ESM has been used for developing machine learning methodologies? If that's the case, in which way?**

  We will skip this part of the sentence, since machine learning is just applied for examination of heat extremes over Europe in historical experiments, but not yet for the assessment of predictions.

- **Sentence in lines 109-110: The phrasing is odd. I would simply say that MPI-ESM has been used to conduct operational decadal climate predictions.**

  Thanks, we will change the sentence accordingly: "MPI-ESM has been also used for the assessment of decadal climate predictions and is used to conduct operational forecasts"

- **Sentence in lines 110-113: This sentence would benefit from some rephrasing too. I suggest simply saying that decadal prediction skill in the model has been shown to arise from near-term memory in the North Atlantic Ocean heat content and from the externally-forced long-term trends. Also, note that "prediction skill" should be written in singular.**

  Thanks, will change the sentence accordingly.

- **Sentence in lines 113-116: Instead of indicating the processes for which the prediction skill has been assessed, it would be more interesting to state those for which the model shows actual skill, and those for which it doesn't.**

  Agreed, we meant the actual skill. The sentence will be changed to: "In addition, actual predictions skill is found…". In addition, we will add a recent publication which demonstrates that summer heat extremes can be predicted by the model (Wallberg et al., 2025).

- **Lines 118-119: The carbon uptake by the ocean as not an Earth System component, it's an Earth system process.**

  True! Will be changed accordingly.

- **Line 172: What is TERRA? You have not properly introduced it.**

  Thanks! TERRA was adopted from the Latin for "earth". This note will be added to the sentence where TERRA first appears.

  For clarity we will also explain the role of TERRA in ICON Land, which has not been stated yet:" ICON Land includes the JSBACH land-surface model developed for predecessors of ICON XPP such as MPI-ESM (Reick et al., 2013, 2021), and other land-surface model such as TERRA (from the Latin for "earth") which is implemented into the operational configuration of ICON NWP"

- **Lines 244 to 246: Node characteristics can largely vary across machines. Could you also indicate the throughput in terms of the number of processors per day (to allow a more direct comparison with other models)?**

  For each node there are two CPUs (64 cores each). We will add a note in the text: "The experiments are run on the CPU-partition of the Levante High-Performance Computing system at the Deutsche Klimarechenzentrum (DKRZ), with each node consisting of 2 CPUs and 128 cores in total."

- **Section 2.3: The manuscript would benefit from additional detail on the tuning procedure. Specifically, it would be helpful to explain how the parameter choices summarized in Table 1 were determined, whether they were based on tailored experiments, and if so, what kind of experiments were conducted.**

  Yes, tailored pre-industrial control experiments with a wider range of parameters were performed to get those optimal parameters and their values in Table 1. Those experiments include a series of coupled model runs to achieve the ideal convection, cloud cover and microphysics parameters for TOA radiation balance and GMT assessment, and then additionally sea-ice and ocean mixing parameters to get a balanced ocean circulation assessment (e.g. AMOC, Labrador Sea freezing and mixed-layer depth). We will include a statement accordingly in the main text:

  " A series of tailored pre-industrial control experiments are employed to find the optimal parameterization values. First, a wider range of convection, microphysics and cloud cover parameters are examined to estimate their impacts on the TOA radiation balance and GMT. Then, with the resulting subset of atmospheric and oceanic parameters the ocean-circulation and sea-ice distributions are adjusted. With the optimized parameters a new spin-up is started. The values of the optimized parameter values are shown in Table 1."

- **Sentence in lines 270-271: I agree that reported TOA values are within the acceptable range and compare well with residual imbalances documented in other CMIP6 models. However, in the last 500 years of the spin-up simulations both the 160/40 and 80/20 configurations exhibit consistently positive and negative TOA imbalances, respectively, without oscillating around zero. This suggests a persistent net energy gain in one case and loss in the other, which may have implications for long-term climate stability and should be acknowledged and discussed in the manuscript.**

  We don't think that a further discussion is needed since such inconsistencies are well-known in coupled models, and – as the reviewer pointed out - the TOA imbalances in the presented configurations are relatively small. Further it is unclear what their net effects on the long-term climate would be, since there are yet unresolved issues such as energy leakages due to missing parametrization, or incomplete atmosphere-ocean coupling, and small but long-term trends in the ocean interior due to the fact that the coupled system would require much longer runtime to reach equilibrium. We think a discussion at this point, would be rather speculative than scientific founded, and would require a proper analysis.

- **Line 304: Could you indicate here and in Table 2 how many members you have run per ensemble?**

  The number of ensemble members are already given in Table 2 in the description of HIST. This information appears perhaps a bit hidden. We will now explain the number of members in the table caption: "For HIST, three ensemble members are performed for the period 1850-2014."

- **Figure 8: In this figure the red color represents the 160/40 configuration and the blue color the 80/20 one, but in Figure 1 is the other way around. I suggest using the same color convention to ease the comparability of the figures.**

  Thanks, we will change the colors accordingly.

- **Line 335: It would be fair to comment that the reference period for PIOMAS corresponds to a much warmer climate than for the preindustrial simulations, which implies that the preindustrial sea ice thickness is expected to be larger.**

  Yes, agreed. We will add a note by the end of this paragraph: "In fact, since the PIOMAS reanalysis depicts the current state of the climate, the preindustrial sea-ice thickness is expected to be larger."

- **Line 344: A peak magnitude should be a flat number, not a range.**

  True. This sentence will be changed to:" For the last 500 years of simulation, the overturning circulations in the Atlantic show values between 14-17 Sv for 80/20 and 16-19 Sv for 160/40 at 26° N at 1000 m depth, which is comparable to the RAPID array (~17 +/-4 Sv, Frajka-Williams et al., 2019)."

- **Line 350: I would explicitly say that you refer to the transport through ocean passages that are important for the climate system.**

  We will add a note where the transports are first mentioned: "… and transport through various ocean passages that are important for various climate sub-systems (table 3)."

- **Figure 5: A key process for the AMOC and decadal variability (and predictability) in the North Atlantic is deep water mixing in the Labrador and Irminger Seas, which is controlled**

**by density stratification. It would be extremely useful to show how they are represented in the two model configurations, given the goal of using them to perform decadal climate predictions.**

We will include a new figure showing the mixed-layer depth for the two configurations and will further add the following text:

" The state of the ocean circulation in the North Atlantic is closely related to the deep-water mixing in the Labrador Sea and Irminger Sea, and at higher latitudes in the Norwegian and Greenland Seas. The deep convection of the Labrador Sea and Irminger Sea can drive the deep-water formation, and are suggested to impact on the AMOC. The mixing in the Norwegian and Greenland Seas contribute to the Arctic overflows and Atlantic bottom water. The mixed-layer depth in March is used here as a proxy for deep-water mixing (Fig. 6). It shows that the 80/20 configuration provides deep mixed layers in the Labrador Sea with maximum values of up to 3000 m. In the Irminger Sea, the mixed-layer depth reaches values of up to 2500 m. The maximum of the deep mixed layers in the 160/40 configuration is shifted to the Irminger Sea and reaches values of about 2500-3000 m. The shift of the maximum values of the mixed-layer depth is closely related to the production of sea ice, which is larger in this configuration compared to the 80/20 configuration (see Fig 1d). The values of mixed-layer depths are generally higher compared to recent climate estimates for which maximum values of ~1000 m in the Labrador Sea and Irminger Sea are suggested (e.g. Königk et al., 2021). Finally, the mixed-layer depths in the Norwegian Sea are similar in both configurations and reach values of up to 3000 m."

[Figure]

Figure 1: The mixed-layer depth in March of CTRL for (a) 160/40 and (b) 80/20. Units are in meters [m].

- **Table 3: The title of the first column is incorrect. It's not an experiment, but an ocean passage that you are listing in the column.**

  Thanks. Typo! Will be changed accordingly.

- **Line 489: I would change "key indicator for" with "critical parameter that determines the".**

  Will be changed accordingly.

- **Sentences in Lines 502-505: The assumption of linearity doesn't seem to hold in the last 130 years of the 80/20 configuration, which has an R square of 0.1 that is most likely not statistically significant. Can you discuss which implications this has for your estimate?**

  The low R-value of the 80/20 experiment is dominated by a group of points near TOA=0. Upon closer examination, we found that this cluster of points corresponds to the model time steps at which technically necessary restarts of the experiment were performed. The configuration for this experiment was extremely unstable, so on-the-fly adjustments were made (e.g., by changing the time steps in the ocean and atmosphere, or the Rayleigh coefficient). These parameter changes were somewhat too strong. This resulted in the model climate adjustment that reflected those changes rather than the climate sensitivity. We repeated the 4xCO2 experiment for the 80/20 configuration with only minor parameter changes. The figure below shows the ECS of this new experiment for all years (150) and the last 130 years. R-value is now 0.53 and ECS=2.49. We will change the figure in the main text accordingly.

[Figure]

Figure 2: The Equilibrium Climate Sensitivity (ECS) as diagnosed from the scatterplot between TOA net radiance and global mean surface temperature anomaly, including a linear regression for 80/20. ECS is estimated from 150 years of the $4xCO_2$ experiments (yellow line) and for the last 130 years (yellow dotted line). R-squared values are included.

- **Line 520: Did you mean to say "principal modes of climate variability"?**

  Yes, will be changed here accordingly and throughout the text.

- **Sentence in lines 528-529: You should specify that this statement refers to the atmosphere.**

  Yes, will be changed accordingly.

- **Sentence in lines 539-540: The phrasing could be improved. I suggest simply saying that you use OLR as it is generally assumed that is a reasonable proxy for deep tropical convection and precipitation.**

Thanks, will be changed accordingly.

- **Paragraphs in lines 541-560 and Figure 12: You discuss (and cite) first the symmetric component, but you show first the antisymmetric one. I would swap the two rows in the figure to follow the order of the discussion. Also, I suggest acknowledging in the text that your comparisons are just visual and do not consider any statistical significance. Indeed, it is unclear to what extent some of the highlighted improvements for the higher resolution happen by chance.**

Thanks, there has been an error when including the figure. In fact, the two rows should be swapped. In addition, in the text the following sentence appeared twice and will be removed: "One exception is the lack of the strong signal of the n=1 WIG waves (cmp. Fig. 12a and Fig. 12b,c)."

Regarding significance we agree and will remove the subordinate clauses in which "improvements" with respect to the configurations appear.

- **Sentence in lines 555-556: You refer to Figure 12d, but should it be to Figure 12a-c, that are the ones corresponding to the antisymmetric component.**

True. We will swap the rows in figure 12, Then 12d becomes 12d-f.

- **Sentence in lines 589-590: I would be more clear if you specify that Tropflux is your reference for evaluation. I didn't notice until I read the caption of the next figure. Also, could you provide some details on that dataset and provide the corresponding reference? It is not a well-known product.**

We will add some details of TropFlux in the figure 13 caption: "As reference in (c-e) TropFlux is used (Praveen Kumar et al., 2012). TropFlux consists of daily and monthly fluxes, SST and wind stress for the tropical region for 30° S to 30° N, and combines ERA-Interim and ISCCP corrected using Global Tropical Moored Buoy Array data from 1979 to present."

- **Line 601: Change "for all" to "for all metrics".**

Will be changed accordingly.

- **Figure 14: It misses a legend in one of the spaghetti plots to explain what each line represents.**

True. We will include a legend in figure 14b.

- **Sentence in lines 606-609: The link between SST and wind stress biases in the western and central Pacific is not entirely clear. Aside from the edges (150°E-160°E and 240°E-270°E), the SST slope is quite similar across both simulations and the reference dataset, which is not the case for wind stress.**

Thanks, we will disentangle this link in the text: "The west-east SST gradient is about 4 °C and the SST slope is close to what is shown in the TropFlux reference. In the western Pacific edge (150°E-160°E), the SST gradients are relatively steep in both configurations. In the eastern Pacific edge (240°E-270°E), the SST gradient reverses in both configurations. The ENSO-related zonal wind stress substantially improves in the higher-resolved configuration

compared to the 160/40 resolution (Fig 14c). In 80/20 the magnitudes are much closer to the reference, and the minimum is shifted eastward closer to what is observed. "

- **Line 618: How do you define this ENSO amplitude?**

Thanks, the ENSO amplitude is defined as the standard deviation of SST anomalies, we will add this definition.

- **Line 622: Can you explain why it is important to evaluate ENSO skewness?**

Evaluating skewness is crucial because many ENSO-related climate impacts (precipitation extremes, teleconnections, drought risk) depend not only on ENSO amplitude but also on whether warm or cold events dominate. Models that simulate ENSO amplitude realistically may still misrepresent skewness, which limits confidence in projections of future ENSO behavior (An & Jin, 2004; Timmermann et al., 2018; Cai et al., 2021). We would prefer not to explain the skewness in such details in the text, since the ENSO paragraph still is extensive and we think that the skewness should be a standard diagnostic for ENSO.

An, S. I., & Jin, F. F. (2004). Nonlinearity and asymmetry of ENSO. Journal of Climate, 17(12), 2399–2412.
Timmermann, A., et al. (2018). El Niño–Southern Oscillation complexity. Nature, 559, 535–545.
Cai, W., et al. (2021). Changing El Niño–Southern Oscillation in a warming climate. Nature Reviews Earth & Environment, 2(9), 628–644.

- **Line 649: Please avoid using the term "significant" in this context as you have not really assessed statistical significance.**

"significant" will be skipped here.

- **Lines 658-660: This sentence could be rephrased for clarity. I interpret that you mean to say that with the cheaper configuration you can more easily explore the space of hyperparameters in your model to identify potential tuning improvements for ENSO representation.**

Thanks! The sentence will be changed to: **"**Thus, the much faster and cheaper configuration can be used to more easily explore the space of hyperparameters to identify potential tuning improvements for ENSO representation."

- **Lines 695-697: I don't think this statement is correct as it is written. The way I understand it, in the extra-tropics, changes in the zonal and meridional jets are closely linked to changes in major modes of climate variability like the NAO, a link that needs to be well represented for the predictability of these modes and their climate impacts.**

Will change the sentence accordingly.

- **Sentence in lines 731-732: I don't understand what you mean to say here.**

These two sentence will be rephrased to: "ERA5 reveals divergence of the E-vector downstream of the maximum zonal wind, which indicates that momentum fluxes are able to force the jets towards the north-eastward direction."

- **Sentence in lines 769-770: For me the most important advantage of showing the wind anomalies is that they better show the downward propagation.**

Thanks for bringing this up. For example, Bushell et al (2020) show that most of the climate models seem to have an eastward time mean wind bias throughout the depth of the QBO, which means that the easterly winds are too weak in the time mean of their multi-model ensemble. This is not the case for these configurations of ICON XPP, especially for the lower to mid equatorial stratosphere. In ICON the easterlies are too strong, hence, ICON XPP is overestimating the easterlies. To account for this bias and focusing stratospheric oscillation itself, the time mean of the zonal mean zonal wind is removed. We will rephrase this paragraph:

"The observed QBO is characterized by descending alternating easterly and westerly jets in the tropical stratosphere and their downward propagation into the troposphere, as shown by the zonally averaged zonal wind (Fig. 17a). In the ICON XPP 160/40 configuration, the descending winds are weakly easterly with a high periodicity of roughly 12 months at 32 km (~ 10 hPa), compared to roughly 28 months in observations (Fig. 17b). For the higher resolution 80/20 a QBO is present and the period increases to 17 months, although the amplitudes still appear smaller than observations (Fig. 17c). A quasi-permanent easterly wind in the lower-to-middle stratosphere is prominent in both resolutions (Fig. 17 b, c). In order to assess the QBO independently from the climatological state, the long-term mean is removed from the QBO time series (Fig. 17e, f). The zonal wind anomalies emphasize that ICON XPP is capable of developing spontaneous QBO phases and their downward propagation (Fig. 17e,f). However, in 160/40 with 90 vertical levels only, the QBO appears disruptive and the downward propagation is not well established (Fig. 17e). The long-term mean equatorial zonal mean wind in the model configurations further exhibit strong easterly winds at an altitude of about 20 km height…"

Bushell, A. C., Anstey, J. A., Butchart, N., Kawatani, Y., Osprey, S., Richter, J. H., Serva, F., Braesicke, P., Cagnazzo, C., Chen, C.-C., Chun, H.-Y., Garcia, R. R., Gray, L. J., Hamilton, K., Kerzenmacher, T., Kim, Y.-H., Lott, F., Mclandress, C., Naoe, H., Scinocca, J., Stockdale, T. N., Watanabe, S., Yoshida, K., & Yukimoto, S. (2020). Evaluation of the Quasi-Biennial Oscillation in global climate models for the SPARC QBO-initiative. *Quarterly Journal of the Royal Meteorological Society*, 1–31. https://doi.org/10.1002/qj.3765

- **Sentence in lines 771-772: For a robust assessment on the simulated QBO periodicity you could perform a spectral analysis of the QBO index for the models and ERA5.**

We thank the reviewer for bringing this up. We agree that showing the equatorial zonal mean wind or the anomaly is perhaps only an exploratory method to assess the quality of the modeled QBO. We further estimate explicitly the QBO amplitude and the period (see figure below). Both quantities are only estimated for the equatorial wind, not the temperature. The amplitude is determined as $\sqrt{2}\,\sigma$, with $\sigma$ is the standard deviation of the de-seasonalized monthly mean zonal wind at each level (Dunkerton and Delisi, 1985). There are some methods for determining the QBO period: using the fast Fourier transform (FFT) or determining the transition times. The FFT method is used here to determine the QBO period.

The figures show the improvements we can get from the higher-resolving configuration. However, we are a bit uncertain to include this figure in the main text, as it would just confirm what can be extracted from the equatorial mean wind cross-sections, which are a common diagnostic though. Instead, we will include a few markers in the text making a statement about the periodicity of each configuration. See revised paragraph in the previous comment.

Dunkerton, T. J. and Delisi, D. P., "Climatology of the Equatorial Lower Stratosphere." Journal of the Atmospheric Sciences, vol. 42, no. 4, pp. 376–396, 1985. doi:10.1175/1520-0469(1985)042<0376:COTELS>2.0.CO;2.

[Figure]

Figure 3: The QBO amplitude (a) and period (b) for ERA5 (black), and the 160/40 configuration (red) and the 80/20 configuration (blue).

- **Sentence in lines 777-779: You cannot really tell if the improvement comes just from the enhanced vertical resolution or from other factors, as there are other things that also change between the two configurations.**

  This sentence will be deleted. However, we note the role of the vertical resolution for the QBO in the next paragraph. In AMIP-like experiments we have tested the number of levels and their vertical grid distances. It turned out that with an L130 vertical grid the model is able to generate internal generated QBO.

- **Sentence in lines 785-786: Could you explain what is different in those experiments with respect to your experiments besides being performed only with the atmospheric component?**

  There are a few differences between the simulations of Niemeier et al. (2024) and those presented in this paper. Niemeier et al used an atmospheric model with a horizontal resolution of 160 km and 130 vertical levels. Therefore, their simulations either share the horizontal resolution with the 160/40 configuration, but not their number of vertical levels (L90), or the vertical resolution with the 80/20 configuration, but not their horizontal resolution. The lower vertical resolution of 90 levels is found too coarse to generate an internally generated QBO. It does, but not so well as presented in Niemeier et al (2024).

  Another important difference is the time step. Niemeier et al used a time step of 360 s for the horizontal grid resolution of 160 km. Simulations for the 160/40 and the 80/20 configuration were performed with a time step of 450 s, which is considerably larger. Coupled tests experiments with the 160/40 configuration revealed a dependency of the representation of the QBO on the timestep. The reason for this is still not fully understood.

The paragraph will be changed accordingly:" In atmosphere-only experiments (160 km, 130 levels), the frequency of the QBO phases has been examined (Niemeier et al., 2024). In these experiments the QBO is well established and benefit from the increase of number of vertical levels. The lower vertical resolution of 90 levels is found too coarse to generate an internally generated QBO. Further, in the atmosphere-only experiments a much smaller time step was used, which seems to further improve the QBO (360 seconds in atmosphere-only experiments compared to 450 seconds in the coupled configurations). However, the reasons for such impact are yet not fully understood. In addition to the QBO, the atmosphere-only experiments reveal a well-represented stratospheric transport. As an example, the transport of the water vapor cloud after the Honga Tonga eruption is found very close to observations (Niemeier et al., 2024)."

- **Sentence in lines 807-811: This comparison is subject to considerable uncertainty, as it is based on counts from a single decade. This is particularly relevant given that the historical simulations do not capture the observed internal variability. To assess whether the model produces a reasonable number of MSSWs, it would be more informative to construct a histogram of event counts across multiple decades, both in the simulations and in ERA5. It would also be useful to examine the seasonal distribution of MSSWs, i.e., the months in which they tend to occur. In your current plot, based on the selected decade, it appears that both models simulate MSSWs earlier in the season compared to ERA5, and the events also seem to be shorter in duration.**

We agree with the reviewer's concerns. A solid classification of the SSWs requires a larger sample size and a more detailed study of both configurations. Both would currently require considerable effort, especially an in-depth analysis of the SSWs of both configurations would increase the scope of this chapter. However, we accept the reviewer's suggestion and plan a separate publication on the Northern Hemisphere stratosphere in ICON XPP. For the presented paper, we will therefore remove the two sections related to the polar vortex and SSW, including Figure 18.

---

## Author Comment (AC2)

We sincerely thank the reviewer for the time he or she spent editing the manuscript. In editing the manuscript, we have invested more in the language. Among other things, several co-authors proofread the text, particularly with regard to readability. The reviewer's comments provided us with a solid foundation for this. We believe this has significantly improved the readability of the manuscript.

Our answers are integrated into the review below.
* * *
Review #2

"This manuscript describes ICON XPP, a new Earth system model configuration that substitutes the previous MPI Earth system model, as part of the ICON framework. ICON XPP is tailored for climate predictions and projections at two available model resolutions, and aims to be used in the CMIP7 suite of models, after further tuning.

**General review:**

The manuscript provides a valuable contribution to the climate modelling community. The manuscript comprehensively documents and evaluates the ICON XPP model components, configuration and its tuning on decadal and seasonal simulation scales, at both model resolutions. Key drivers of climate prediction and projection are presented and assessed, comparing against other well-established models and ERA5 observational datasets. Assessment and comparison of the models' ability to simulate key measures of climate state, such as climate sensitivity and also key dynamical processes clearly demonstrates where the model performs well and where further improvement could be beneficial. Overall, the manuscript is generally well-written, and the authors effectively present the evaluation of this new Earth system model using key metrics.

I therefore find this paper suitable for publication in GMD, after the author's response/ corrections to a few minor revisions below:

**General:**

- **Overall, the manuscript is well-written, however I believe the manuscript could be condensed as there are a number of lengthy sentences/ paragraphs throughout, that should be more concise for clarity and understanding.**
  We agree. As suggested by the reviewer, we will revise shorten the abstract and restructure the introduction. See individual comments below. We also will check the entire text for clarity.

- **There are also a number of spelling and grammatical mistakes throughout, I have highlighted some of these below, but I recommend a thorough read through to correct these.**
  We will read the text thoroughly with a focus on spell-check and grammatical mistakes. In particular, several co-authors will have a cross-check with regard to the readability of the manuscript.

**General Abstract:**

- **The abstract is too long, in my opinion. Ideally the abstract could be condensed into a more concise version, with only key results and information presented. There is some information I believe could be removed/ sentences condensed to reduce the size.**

Agreed. The abstract will be reduced by a couple of sentences and subordinate clauses. For example, we have identified the role of the model initiative from which ICON XPP is results, and some very specific text passages about the basic assessment, ENSO and the stratospheric dynamics as clear candidates for cut off.

**Introduction general:**

- **While the introduction provides a good presentation of background information required for understanding and guiding the reader through the manuscript, I find the could be structured improved. There is some repetition in some paragraphs, or others seem to continue discussing a point made in previous paragraphs throughout all paragraphs. I would recommend some minor restructuring of the introduction, so it easily flows and provides a succinct background to the great main body of the manuscript.**

  **e.g Line 82:** *"Here, we present ICON XPP, from the design of the configurations to a first evaluation of the Earth System state based on the CMIP DECK (Diagnostic, Evaluation and Characterization of Klima) experimentation framework (Eyring et al., 2016)."*
  **This reads as if it should be placed at the end of the introduction? But this could be a personal opinion.**
  Yes, we agree. We will sharpen the individual paragraphs by shuffling a couple of sentences. For example, the sentence including "…new modeling initiative…" will better be placed in the 2$^{nd}$ paragraph, where the model structure is described. Also, the CMIP DECK experiment is better placed in the paragraph about climate projections. The climate projection paragraph will be shifted upward, right after the predictions, and fits much clearer the textual order (prediction, projections, research).

  In addition, some subordinate clauses will be deleted. On example is – as already mentioned by the reviewer – the relative complex last sentence in the first paragraph, which will be rephrased to: "Special attention is given to monitoring certain aspects of the tropical and extra-tropical mean climate, including key modes of variability. "

**Minor corrections:**

- **Line 47: "*At regional scale*". Should this read: "At a regional scale"?**
  Will be changed accordingly.

- **Figure 1 and Figure 8: The colours used for each model resolution is switched in these figures, ensuring all figures have a consistent colour scheme would improve clarity.**
  Yes, sorry! The colors in figure 8 will be synchronized with figure 1.

- **Lines 76-78: *"Since 2020, a new modelling initiative integrating numerical weather forecast, climate predictions and climate projections based on the ICON framework (Müller et al., 2025)."* This sentence seems unfinished?**
  This sentence will be rephrased and shifted to the next paragraph. It will read: "Since 2020, a new modeling initiative integrates numerical weather forecast, climate predictions and climate projections based on the ICON framework into a single model system (Müller et al., 2025)."

- **Line 79: '-,' - The comma isn't needed.**
  Agreed.

- **Line 80:** *"ICON XPP will be the baseline for next generation climate predictions …"* Should be: **"ICON XPP will be the baseline for the next generation of climate predictions…?**
  Will be changed accordingly.

- **Lines 84-87:** *"Special attention is given to monitoring certain aspects of the tropical and extra-tropical mean climate, and the stratosphere, including key modes of variability and their predictability, such as the El Niño/Southern Oscillation (ENSO), or the North Atlantic Oscillation (NAO)."* **- This sentence is slightly verbose and clunky.**
  The sentence will be changed to: **"**Special attention is given to monitoring certain aspects of the tropical and extra-tropical mean climate, and key modes of variability."

- **Line 88:** *"for the individual components"* **- Leaves the reader wondering what the individual components are. Could point to a table or list of the components, or leave the "individual components" part out?**
  This sentence will be changed to:" ICON XPP builds upon accomplishments of previous ICON initiatives with regard to the climate sub-components."

- **Line 107: Needs to be clearer**
  The part "continental-scale temperature and precipitation patterns" will be deleted.

- **Lines 107-108:** *"and it is recently for machine learning methodologies to assess…"* **- Should read, "and it has recently been used for machine learning methodologies to assess…"? If not, this sentence should be re-phrased.**
  The subordinate clause about machine learning will be deleted.

- **Line 109:** *"MPI-ESM has been also used for decadal climate"* **- Should read, "MPI-ESM has also been used for decadal climate".**
  Will be changed accordingly.

- **Line 172: "TERRA" isn't defined? A definition or acronym description here would be useful.**
  Thanks! TERRA was adopted from the Latin for "earth". This note will be added to the sentence where TERRA first appears.

  For clarity we will also explain the role of TERRA in ICON Land, which has not been stated yet:" ICON Land includes the JSBACH land-surface model developed for predecessors of ICON XPP such as MPI-ESM (Reick et al., 2013, 2021), and other land-surface model such as TERRA (from the Latin for "earth"), which is implemented into the operational configuration of ICON NWP"

- **Line 264:** *"2 metre"* **- Should it read "2 meters"?**
  Will be changed accordingly.

- **Lines 262-265:** *"The targets mainly consider the thermodynamic state of the atmosphere - depicted by the top-of-atmosphere radiation balance and global-mean temperature at 2 metre - and the ocean-cryosphere - by the strength of the Atlantic meridional overturning circulation (AMOC) and sea-ice properties"* **- A long sentence, could split into 2? The whole first paragraph of the 2.3 Tuning section could be re-worded for flow and clarity.**

The sentence will be rephrased: "The targets mainly consider the top-of-atmosphere (TOA) radiation balance and global-mean temperature at 2 meters (GMT) - and the strength of the Atlantic meridional overturning circulation (AMOC) and sea-ice properties. "

- **Line 276: "@26° N" – is this the correct symbol?**
  Will be changed to "at 26° N"

- **Line 277: "*a small trend remains for the AMOC at the end of the simulation.*" - Should this read 'by the end of simulation", instead of "at the end of simulation"? This would provide more clarity for me, implying a trend still remains by the end of the simulation? The authors could also elaborate on this trend more? A negative trend/ the AMOC weakens?**
  Thanks! The sentence will be changed to: In 160/40 a small negative trend of the AMOC at the end of the simulation. **"**

- **Line 284: "sediment,." – should be "sediment."? No need for extra comma.**
  Will be changed accordingly.

- **Table 1: Could the sea ice parameters have units? i.e Sea ice melting 0.25, is this millionkm²? Or is there a reason no units are provided? Some more information could be provided on the tuning process for the reader to understand how each value was reached.**
  The sea-ice parameters are dimensionless. We will add a statement of the units and whether they are dimensionless in the figure caption.

  In addition, to get the optimized values we performed a series of tailored experiment targeting the climate benchmarks. We will include a statement accordingly in the main text to make this a bit clearer:

  " A series of tailored pre-industrial control experiments are employed to find the optimal parameterization values. First, a wider range of convection, microphysics and cloud cover parameters are examined to estimate their impacts on the TOA radiation balance and GMT. Then, with the resulting subset of atmospheric and oceanic parameters the ocean-circulation and sea-ice distributions are adjusted. With the optimized parameters a new spin-up is started. The values of the optimized parameter values are shown in Table 1."

- **Line 307: "*The climate sensitivity is estimated by 1%...*", should read "The climate sensitivity is estimated by a 1%..."?**
  Will be changed accordingly.

- **Line 324: Again, I may have missed it but "AMIP-type", could be defined or the acronym breakdown provided.**
  Will replace "AMIP-type" by "atmosphere only"

- **Line 328: "…for 160/40 and ~1.7 °C 80/20,…", missing a 'for'? should read "…for 160/40 and ~1.7 °C for 80/20,…"**
  Thanks. Will be changed accordingly.

- **Line 330: "*The sea-ice reveal reasonable…*", missing 'simulations'? Should read: "The sea-ice simulations reveal"?**
  Thanks. Will be changed accordingly.

- **Line 330: "peak season"/ "minimum season" could be defined? As the terms "peak season"/ "winter season/ growth season" etc are used interchangeably. Could be useful to provide the months referred to in brackets.**
  We will replace "peak" by "winter" and "minimum" by "summer". "Winter" and "summer" are defined in the figure caption.

- **Line 344: *"peak magnitudes of ~15-20 Sv at 26° N at 1000m depth,…"*, should it read "and 1000m depth…"? In addition, should the peak magnitude be a range, or a single number?**
  Yes, thanks. We will change this sentence to: **"**For the last 500 years of simulation, the overturning circulations in the Atlantic at 26° N and 1000 m depth show values between 14-17 Sv for 80/20 and 16-19 Sv for 160/40,…"

- **Table 3: misaligned data in the observations column/ Denmark Strait row?**
  Will be changed accordingly.

- **Line 423: "a *small ensemble of three…"* – I would be interested to know why three were generated? Was this a computational constraint, or something else? An explanation in-text could be useful?**
  No, unfortunately not. The number is simply a guess to see whether the experiments roughly do similar by changing the initial conditions. Since this is based more on "modeler's experience" rather than "computational constraints" we would like to avoid a discussion in the text.

- **Line 437: "*The causes are currently unclear, and further investigations are in progress."* Is there any literature available yet? Is it possible to cite these further investigations?**
  We modified this sentence and provide references to support the text, the new lines are:
  "Although the causes of the double-ITCZ are currently unclear, some models have modified the clouds microphysics, vertical entrainment rates, convection schemes or the atmospheric energy balance to reduce this feature (e.g., Ma et al., 2023; Ren and Zhou, 2024); however, no generalized modification can be applied to all models"

  Ma, X., S. Zhao, H. Zhang, and W. Wang, 2023. The double-ITCZ problem in CMIP6 and the influences of deep convection and model resolution. International Journal of Climatology, 43, 2369-2390, doi: 10.1002/joc.7980.
  Ren, Z., and T. Zhou, 2024. Understanding the alleviation of "double-ITCZ" bias in CMIP6 models from the perspective of atmospheric energy balance. *Climate Dynamics*, 62, 6819-6839, doi: 10.1007/s00382-024-07238-7.

- **Line 448: *"Smaller errors"* – could be made clearer? "Smaller margin of error"?**
  We will change this sentence to: "Smaller root mean squared errors (RMSE) are found for many dynamical and thermodynamical quantities by increasing the resolution from the 160/40 to the 80/20 configuration."

- **Table 4: It would be nice to see Table 4 placed nearer to where comparison to observations is mentioned, if possible.**
  Will be changed accordingly.

- **Line 577: Could define the *"cold tongue bias"* better, this sentence could also benefit from some citation.**
  Thanks, we will add the following: " The cold tongue bias refers to the excessive cooling along the equatorial Pacific, a common systematic error in climate models (Li & Xie, 2014)."
  Li, G., & Xie, S.-P. (2014). Tropical biases in CMIP5 multimodel ensemble: The excessive equatorial Pacific cold tongue and double ITCZ problems.

- **Line 588: *"Nino3.4-related"*, could you introduce this? A definition or explanation would be useful here.**
  Thanks, this sentence now starts with: "A regression of SST anomalies to the Nino3.4 index" to make the Nino3.4 region more explicit. Further the Nino3.4 index will be defined in the figure caption.

- **Line 590: *"As many coupled models…"* – should read "As in many coupled models…"?**
  Will be changed accordingly.

- **Line 691: *"Among other…"* – should read "Among others…"?**
  Will be changed accordingly.

- **Line 698: *"… and meanwhile…"*, do you need both?**
  In fact, not both are needed. The sentence will be changed to: "…and seasonal and decadal prediction skill of the NAO is established …"

- **Line 716: Full stop missing after citation (Müller et al., 2018)**
  Will be changed accordingly.

- **Line 737: *"well-behaviour"*, is this the appropriate word? Are you referring to good performance?**
  Thanks. Will be changed to "good performance"

- **Line 786: Brackets are needed around the citation (Niemeier et al., 2024), or sentence needs re-phrasing if not.**
  Will be changed accordingly.

- **Line 795: *"However, several factors – among others volcanic eruptions…"*, grammatical errors here/ more punctuation needed.**
  We will remove the two sections on the polar vortex and SSWs (including Figure 18), as it would require more effort to elaborate on the SSWs in the necessary depth. In response to the first reviewer, it would be necessary to consider several decades to quantify the necessary variability of the SSWs, as well as a more in-depth analysis beyond the presentation of the time series. Both would require currently considerable effort, especially an in-depth analysis of the SSWs of both configurations would increase the scope of this chapter.

- **Line 806: The sentence would read better if it were *"winter northern hemisphere"* not "hemispheric"? Punctuation is also missing from this sentence: *"shows, for example, …".***
  Please see comment above.

- **Line 812:** *"exhibits comparable variations as in observations",* **this sentence could be much shorter e.g "comparable to observations".**
  Please see comment above.

---

## Author Response (AR1)

We sincerely thank the reviewers for the time he or she spent editing the manuscript. In editing the manuscript, we have invested more in the language. Among other things, several co-authors proofread the text, particularly with regard to readability. The reviewer's comments provided us with a solid foundation for this. We believe this has significantly improved the readability of the manuscript.

Our answers are integrated into the review below.

In addition to the reviewer's comments the revised manuscript contains a few more changes.

- As noticed by the editorial support team the colored tables are changed to meet the journal criteria.
- The sea-ice description has been wrong and changed to "Sea-ice thermodynamic is calculated in the atmospheric part and uses the zero-layer model (Mironov, et. al, 2012, Semtner Jr., 1976). Melting potential and conductive heat flux are passed to the ocean component by use of the YAC coupler." (I193ff)
- Further we noticed two bugs in the ENSO diagnostics. First, the teleconnection diagnostics considered SST regression to the ENSO index instead of T2m regression to the ENSO index. Using T2m leads to a weaker performance in the overall ICON XPP teleconnection score. The text is changed accordingly. Second, for the CMIP reference categories in Fig 15a, we mistakenly included all individual metric values instead of their mean when plotting the box plot. This leads to narrower box plots for the CMIP reference.
* * *
**Review #1**

"This manuscript presents ICON XPP, the first version of a new climate model specifically tailored for climate prediction applications at two alternative model resolutions.

Both the model documentation and the evaluation of the two resolution configurations are comprehensive, spanning seasonal to decadal scales and covering important drivers of predictability such as the ocean and atmospheric circulations, the stratosphere and key modes of internal climate variability. The authors employ a wide range of diagnostics and benchmark against well-established datasets, demonstrating the model's strengths and identifying areas for improvement.

I find the paper to be of interest and suitable for publication in GMD, pending the authors' response to a few clarifications and minor comments listed below. "

The manuscript is generally clear but presents frequent typos and grammatical errors. I
recommend a thorough proofreading to improve its readability.

Thanks a lot for pointing at this. We have made a thorough check, for example proofreading by a few co-authors. In relation to reviewer 2, we also sharpen the abstract and restructure the introduction.

Sentence in lines 105-109: The final part of the sentence seems to be incomplete. Did you
mean to say that MPI-ESM has been used for developing machine learning methodologies?
If that's the case, in which way?

We skipped this part of the sentence, since machine learning is just applied for examination of heat extremes over Europe in historical experiments, but not yet for the assessment of predictions.

• Sentence in lines 109-110: The phrasing is odd. I would simply say that MPI-ESM has been used to conduct operational decadal climate predictions.

Thanks, we changed the sentence accordingly: "MPI-ESM has been also used for the assessment of decadal climate predictions and is used to conduct operational forecasts" (I92-93)

 Sentence in lines 110-113: This sentence would benefit from some rephrasing too. I suggest simply saying that decadal prediction skill in the model has been shown to arise from nearterm memory in the North Atlantic Ocean heat content and from the externally-forced long-term trends. Also, note that "prediction skill" should be written in singular.

Thanks, we changed the sentence accordingly. (196-98)

Sentence in lines 113-116: Instead of indicating the processes for which the prediction skill
has been assessed, it would be more interesting to state those for which the model shows
actual skill, and those for which it doesn't.

Agreed, we meant the actual skill. The sentence is changed to: "In addition, actual predictions skill is found...". In addition, we will add a recent publication which demonstrates that summer heat extremes can be predicted by the model (Wallberg et al., 2025). (196-101)

• Lines 118-119: The carbon uptake by the ocean as not an Earth System component, it's an Earth system process.

True! Changed accordingly.

Line 172: What is TERRA? You have not properly introduced it.

Thanks! TERRA was adopted from the Latin for "earth". This note is added to the sentence where TERRA first appears. (I145-146)

For clarity we also explain the role of TERRA in ICON Land, which has not been stated yet: "ICON Land includes the JSBACH land-surface model developed for predecessors of ICON XPP such as MPI-ESM (Reick et al., 2013, 2021), and other land-surface model such as TERRA (from the Latin for "earth") which is implemented into the operational configuration of ICON NWP" (I143ff)

 Lines 244 to 246: Node characteristics can largely vary across machines. Could you also indicate the throughput in terms of the number of processors per day (to allow a more direct comparison with other models)?

For each node there are two CPUs (64 cores each). We added a note in the text: "The experiments are run on the CPU-partition of the Levante High-Performance Computing system at the Deutsche Klimarechenzentrum (DKRZ), with each node consisting of 2 CPUs and 128 cores in total." (I228ff)

Section 2.3: The manuscript would benefit from additional detail on the tuning procedure.
 Specifically, it would be helpful to explain how the parameter choices summarized in Table 1 were determined, whether they were based on tailored experiments, and if so, what kind of experiments were conducted.

Yes, tailored pre-industrial control experiments with a wider range of parameters were performed to get those optimal parameters and their values in Table 1. Those experiments include a series of coupled model runs to achieve the ideal convection, cloud cover and microphysics parameters for TOA radiation balance and GMT assessment, and then additionally sea-ice and ocean mixing parameters to get a balanced ocean circulation assessment (e.g. AMOC, Labrador Sea freezing and mixed-layer depth). We included a statement accordingly in the main text:

- "A series of tailored pre-industrial control experiments are employed to find the optimal parameterization values. First, a wider range of convection, microphysics and cloud cover parameters are examined to estimate their impacts on the TOA radiation balance and GMT. Then, with the resulting subset of atmospheric and oceanic parameters the ocean-circulation and sea-ice distributions are adjusted. With the optimized parameters a new spin-up is started. The values of the optimized parameter values are shown in Table 1." (1252ff)
- Sentence in lines 270-271: I agree that reported TOA values are within the acceptable range and compare well with residual imbalances documented in other CMIP6 models. However, in the last 500 years of the spin-up simulations both the 160/40 and 80/20 configurations exhibit consistently positive and negative TOA imbalances, respectively, without oscillating around zero. This suggests a persistent net energy gain in one case and loss in the other, which may have implications for long-term climate stability and should be acknowledged and discussed in the manuscript.

We don't think that a further discussion is needed since such inconsistencies are well-known in coupled models, and – as the reviewer pointed out - the TOA imbalances in the presented configurations are relatively small. Further it is unclear what their net effects on the longterm climate would be, since there are yet unresolved issues such as energy leakages due to missing parametrization, or incomplete atmosphere-ocean coupling, and small but long-term trends in the ocean interior due to the fact that the coupled system would require much longer runtime to reach equilibrium. We think a discussion at this point, would be rather speculative than scientific founded, and would require a proper analysis.

 Line 304: Could you indicate here and in Table 2 how many members you have run per ensemble?

The number of ensemble members are already given in Table 2 in the description of HIST. This information appears perhaps a bit hidden. We now explain the number of members in the table caption: "For HIST, three ensemble members are performed for the period 1850-2014."

 Figure 8: In this figure the red color represents the 160/40 configuration and the blue color the 80/20 one, but in Figure 1 is the other way around. I suggest using the same color convention to ease the comparability of the figures.

Thanks, we changed the colors accordingly.

 Line 335: It would be fair to comment that the reference period for PIOMAS corresponds to a much warmer climate than for the preindustrial simulations, which implies that the preindustrial sea ice thickness is expected to be larger.

Yes, agreed. We added a note by the end of this paragraph: "In fact, since the PIOMAS reanalysis depicts the current state of the climate, the preindustrial sea-ice thickness is expected to be larger." (I332ff)

• Line 344: A peak magnitude should be a flat number, not a range.

True. This sentence is changed to:" For the last 500 years of simulation, the overturning circulations in the Atlantic show values between 14-17 Sv for 80/20 and 16-19 Sv for 160/40 at  $26^{\circ}$  N at 1000 m depth, which is comparable to the RAPID array (~17 +/-4 Sv, Frajka-Williams et al., 2019)." (1337ff)

• Line 350: I would explicitly say that you refer to the transport through ocean passages that are important for the climate system.

We added a note where the transports are first mentioned: "... and transport through various ocean passages that are important for various climate sub-systems (table 3)."

Figure 5: A key process for the AMOC and decadal variability (and predictability) in the
North Atlantic is deep water mixing in the Labrador and Irminger Seas, which is controlled
by density stratification. It would be extremely useful to show how they are represented in
the two model configurations, given the goal of using them to perform decadal climate
predictions.

We included a new figure showing the mixed-layer depth for the two configurations and will further add the following text:

"The state of the ocean circulation in the North Atlantic is closely related to the deep-water mixing in the Labrador Sea and Irminger Sea, and at higher latitudes in the Norwegian and Greenland Seas. The deep convection of the Labrador Sea and Irminger Sea can drive the deep-water formation, and are suggested to impact on the AMOC. The mixing in the Norwegian and Greenland Seas contribute to the Arctic overflows and Atlantic bottom water. The mixed-layer depth in March is used here as a proxy for deep-water mixing (Fig. 6). It

shows that the 80/20 configuration provides deep mixed layers in the Labrador Sea with maximum values of up to 3000 m. In the Irminger Sea, the mixed-layer depth reaches values of up to 2500 m. The maximum of the deep mixed layers in the 160/40 configuration is shifted to the Irminger Sea and reaches values of about 2500-3000 m. The shift of the maximum values of the mixed-layer depth is closely related to the production of sea ice, which is larger in this configuration compared to the 80/20 configuration (see Fig 1d). The values of mixed-layer depths are generally higher compared to recent climate estimates for which maximum values of ~1000 m in the Labrador Sea and Irminger Sea are suggested (e.g. Königk et al., 2021). Finally, the mixed-layer depths in the Norwegian Sea are similar in both configurations and reach values of up to 3000 m." (I372ff)

Figure 1: The mixed-layer depth in March of CTRL for (a) 160/40 and (b) 80/20. Units are in meters [m].

• Table 3: The title of the first column is incorrect. It's not an experiment, but an ocean passage that you are listing in the column.

Thanks. Typo! Changed accordingly.

- Line 489: I would change "key indicator for" with "critical parameter that determines the".
   Changed accordingly.
- Sentences in Lines 502-505: The assumption of linearity doesn't seem to hold in the last 130 years of the 80/20 configuration, which has an R square of 0.1 that is most likely not statistically significant. Can you discuss which implications this has for your estimate?

The low R-value of the 80/20 experiment is dominated by a group of points near TOA=0. Upon closer examination, we found that this cluster of points corresponds to the model time steps at which technically necessary restarts of the experiment were performed. The configuration for this experiment was extremely unstable, so on-the-fly adjustments were made (e.g., by changing the time steps in the ocean and atmosphere, or the Rayleigh coefficient). These parameter changes were somewhat too strong. This resulted in the model climate adjustment that reflected those changes rather than the climate sensitivity. We repeated the 4xCO2 experiment for the 80/20 configuration with only minor parameter changes. The figure below shows the ECS of this new experiment for all years (150) and the last 130 years. R-value is now 0.53 and ECS=2.49. We changed the figure (now figure) 12 in the main text accordingly.

Figure 2: The Equilibrium Climate Sensitivity (ECS) as diagnosed from the scatterplot between TOA net radiance and global mean surface temperature anomaly, including a linear regression for 80/20. ECS is estimated from 150 years of the  $4xCO_2$  experiments (black), the first 20 years (blue) and for the last 130 years (yellow). R-squared values are included.

- Line 520: Did you mean to say "principal modes of climate variability"?
   Yes, changed here accordingly and throughout the text.
- Sentence in lines 528-529: You should specify that this statement refers to the atmosphere.
   Agreed, changed accordingly.
- Sentence in lines 539-540: The phrasing could be improved. I suggest simply saying that you
  use OLR as it is generally assumed that is a reasonable proxy for deep tropical convection
  and precipitation.

Thanks, changed accordingly. (I558-559)

Paragraphs in lines 541-560 and Figure 12: You discuss (and cite) first the symmetric
component, but you show first the antisymmetric one. I would swap the two rows in the
figure to follow the order of the discussion. Also, I suggest acknowledging in the text that
your comparisons are just visual and do not consider any statistical significance. Indeed, it
is unclear to what extent some of the highlighted improvements for the higher resolution
happen by chance.

Thanks, there has been an error when including the figure. Figure 12 is now Figure 13. In fact, the two rows were swapped. In addition, in the text the following sentence appeared twice and is removed: "One exception is the lack of the strong signal of the n=1 WIG waves (cmp. Fig. 13a and Fig. 13b,c)."

Regarding significance we agree and removed the subordinate clauses in which "improvements" with respect to the configurations appear.

• Sentence in lines 555-556: You refer to Figure 12d, but should it be to Figure 12a-c, that are the ones corresponding to the antisymmetric component.

True. Figure 12 now becomes Figure 13. We swapped the rows in figure 13, Then 13d becomes 13d-f.

Sentence in lines 589-590: I would be more clear if you specify that Tropflux is your
reference for evaluation. I didn't notice until I read the caption of the next figure. Also,
could you provide some details on that dataset and provide the corresponding reference? It
is not a well-known product.

We added some details of TropFlux in the figure caption (figure 13 now becomes figure 14): "As reference in (c-e) TropFlux is used (Praveen Kumar et al., 2012). TropFlux consists of daily and monthly fluxes, SST and wind stress for the tropical region for 30° S to 30° N, and combines ERA-Interim and ISCCP corrected using Global Tropical Moored Buoy Array data from 1979 to present."

Line 601: Change "for all" to "for all metrics".

Changed accordingly.

 Figure 14: It misses a legend in one of the spaghetti plots to explain what each line represents.

True. We included a legend in the figure. Figure 14 now becomes figure 15.

 Sentence in lines 606-609: The link between SST and wind stress biases in the western and central Pacific is not entirely clear. Aside from the edges (150°E-160°E and 240°E-270°E), the SST slope is quite similar across both simulations and the reference dataset, which is not the case for wind stress.

Thanks, we disentangled this link in the text: "The west-east SST gradient is about 4 °C and the SST slope is close to what is shown in the TropFlux reference. In the western Pacific edge (150°E-160°E), the SST gradients are relatively steep in both configurations. In the eastern Pacific edge (240°E-270°E), the SST gradient reverses in both configurations. The ENSO-related zonal wind stress substantially improves in the higher-resolved configuration compared to the 160/40 resolution (Fig 15c). In 80/20 the magnitudes are much closer to the reference, and the minimum is shifted eastward closer to what is observed. "(1622ff)

Line 618: How do you define this ENSO amplitude?

Thanks, the ENSO amplitude is defined as the standard deviation of SST anomalies. This definition is now added. (I636)

Line 622: Can you explain why it is important to evaluate ENSO skewness?

Evaluating skewness is crucial because many ENSO-related climate impacts (precipitation extremes, teleconnections, drought risk) depend not only on ENSO amplitude but also on whether warm or cold events dominate. Models that simulate ENSO amplitude realistically may still misrepresent skewness, which limits confidence in projections of future ENSO behavior (An & Jin, 2004; Timmermann et al., 2018; Cai et al., 2021). We would prefer not to explain the skewness in such details in the text, since the ENSO paragraph still is extensive and we think that the skewness should be a standard diagnostic for ENSO.

An, S. I., & Jin, F. F. (2004). Nonlinearity and asymmetry of ENSO. Journal of Climate, 17(12), 2399–2412.

Timmermann, A., et al. (2018). El Niño–Southern Oscillation complexity. Nature, 559, 535–545.

Cai, W., et al. (2021). Changing El Niño—Southern Oscillation in a warming climate. Nature Reviews Earth & Environment, 2(9), 628–644.

• Line 649: Please avoid using the term "significant" in this context as you have not really assessed statistical significance.

The expression "significant" is skipped here.

Lines 658-660: This sentence could be rephrased for clarity. I interpret that you mean to say
that with the cheaper configuration you can more easily explore the space of
hyperparameters in your model to identify potential tuning improvements for ENSO
representation.

Thanks! The sentence is changed to: "Thus, the much faster and cheaper configuration can be used to more easily explore the space of hyperparameters to identify potential tuning improvements for ENSO representation." (I674ff)

Lines 695-697: I don't think this statement is correct as it is written. The way I understand
it, in the extra-tropics, changes in the zonal and meridional jets are closely linked to
changes in major modes of climate variability like the NAO, a link that needs to be well
represented for the predictability of these modes and their climate impacts.

Changed the sentence accordingly. (I715ff)

Sentence in lines 731-732: I don't understand what you mean to say here.

These two sentence are rephrased to: "ERA5 reveals divergence of the E-vector downstream of the maximum zonal wind, which indicates that momentum fluxes are able to force the jets towards the north-eastward direction." (I751ff)

 Sentence in lines 769-770: For me the most important advantage of showing the wind anomalies is that they better show the downward propagation.

Thanks for bringing this up. For example, Bushell et al (2020) show that most of the climate models seem to have an eastward time mean wind bias throughout the depth of the QBO, which means that the easterly winds are too weak in the time mean of their multi-model ensemble. This is not the case for these configurations of ICON XPP, especially for the lower to mid equatorial stratosphere. In ICON the easterlies are too strong, hence, ICON XPP is overestimating the easterlies. To account for this bias and focusing stratospheric oscillation itself, the time mean of the zonal mean zonal wind is removed. We rephrased this paragraph:

"The observed QBO is characterized by descending alternating easterly and westerly jets in the tropical stratosphere and their downward propagation into the troposphere, as shown by the zonally averaged zonal wind (Fig. 18a). In the ICON XPP 160/40 configuration, the descending winds are weakly easterly with a high periodicity of roughly 12 months at 32 km (~ 10 hPa), compared to roughly 28 months in observations (Fig. 18b). For the higher resolution 80/20 a QBO is present and the period increases to 17 months, although the amplitudes still appear smaller than observations (Fig. 18c). A quasi-permanent easterly wind in the lower-to-middle stratosphere is prominent in both resolutions (Fig. 18b, c). In order to assess the QBO independently from the climatological state, the long-term mean is removed from the QBO time series (Fig. 18e, f). The zonal wind anomalies emphasize that ICON XPP is capable of developing spontaneous QBO phases and their downward propagation (Fig. 18e,

f). However, in 160/40 with 90 vertical levels only, the QBO appears disruptive and the downward propagation is not well established (Fig. 18e). The long-term mean equatorial zonal mean wind in the model configurations further exhibit strong easterly winds at an altitude of about 20 km height..." (I786)

Bushell, A. C., Anstey, J. A., Butchart, N., Kawatani, Y., Osprey, S., Richter, J. H., Serva, F., Braesicke, P., Cagnazzo, C., Chen, C.-C., Chun, H.-Y., Garcia, R. R., Gray, L. J., Hamilton, K., Kerzenmacher, T., Kim, Y.-H., Lott, F., Mclandress, C., Naoe, H., Scinocca, J., Stockdale, T. N., Watanabe, S., Yoshida, K., & Yukimoto, S. (2020). Evaluation of the Quasi-Biennial Oscillation in global climate models for the SPARC QBO-initiative. *Quarterly Journal of the Royal Meteorological Society*, 1–31. <a href="https://doi.org/10.1002/qj.3765">https://doi.org/10.1002/qj.3765</a>

 Sentence in lines 771-772: For a robust assessment on the simulated QBO periodicity you could perform a spectral analysis of the QBO index for the models and ERA5.

We thank the reviewer for bringing this up. We agree that showing the equatorial zonal mean wind or the anomaly is perhaps only an exploratory method to assess the quality of the modeled QBO. We further estimate explicitly the QBO amplitude and the period (see figure below). Both quantities are only estimated for the equatorial wind, not the temperature. The amplitude is determined as  $V2\sigma$ , with  $\sigma$  is the standard deviation of the de-seasonalized monthly mean zonal wind at each level (Dunkerton and Delisi, 1985). There are some methods for determining the QBO period: using the fast Fourier transform (FFT) or determining the transition times. The FFT method is used here to determine the QBO period.

The figures show the improvements we can get from the higher-resolving configuration. However, we are a bit uncertain to include this figure in the main text, as it would just confirm what can be extracted from the equatorial mean wind cross-sections, which are a common diagnostic though. Instead, we included a few markers in the text making a statement about the periodicity of each configuration. See revised paragraph in the previous comment.

Dunkerton, T. J. and Delisi, D. P., "Climatology of the Equatorial Lower Stratosphere." Journal of the Atmospheric Sciences, vol. 42, no. 4, pp. 376–396, 1985. doi:10.1175/1520-0469(1985)042<0376:COTELS>2.0.CO;2.

Figure 3: The QBO amplitude (a) and period (b) for ERA5 (black), and the 160/40 configuration (red) and the 80/20 configuration (blue).

 Sentence in lines 777-779: You cannot really tell if the improvement comes just from the enhanced vertical resolution or from other factors, as there are other things that also change between the two configurations.

This sentence was deleted. However, we note the role of the vertical resolution for the QBO in the next paragraph. In AMIP-like experiments we have tested the number of levels and their vertical grid distances. It turned out that with an L130 vertical grid the model is able to generate internal generated QBO. (I304ff)

 Sentence in lines 785-786: Could you explain what is different in those experiments with respect to your experiments besides being performed only with the atmospheric component?

There are a few differences between the simulations of Niemeier et al. (2024) and those presented in this paper. Niemeier et al used an atmospheric model with a horizontal resolution of 160 km and 130 vertical levels. Therefore, their simulations either share the horizontal resolution with the 160/40 configuration, but not their number of vertical levels (L90), or the vertical resolution with the 80/20 configuration, but not their horizontal resolution. The lower vertical resolution of 90 levels is found too coarse to generate an internally generated QBO. It does, but not so well as presented in Niemeier et al (2024).

Another important difference is the time step. Niemeier et al used a time step of 360 s for the horizontal grid resolution of 160 km. Simulations for the 160/40 and the 80/20 configuration were performed with a time step of 450 s, which is considerably larger. Coupled tests experiments with the 160/40 configuration revealed a dependency of the representation of the QBO on the timestep. The reason for this is still not fully understood.

The paragraph is changed accordingly:" In atmosphere-only experiments (160 km, 130 levels), the frequency of the QBO phases has been examined (Niemeier et al., 2024). In these experiments the QBO is well established and benefit from the increase of number of vertical levels. The lower vertical resolution of 90 levels is found too coarse to generate an internally generated QBO. Further, in the atmosphere-only experiments a much smaller time step was used, which seems to further improve the QBO (360 seconds in atmosphere-only experiments compared to 450 seconds in the coupled configurations). However, the reasons for such impact are yet not fully understood. In addition to the QBO, the atmosphere-only experiments reveal a well-represented stratospheric transport. As an example, the transport of the water vapor cloud after the Honga Tonga eruption is found very close to observations (Niemeier et al., 2024)." (I304ff)

Sentence in lines 807-811: This comparison is subject to considerable uncertainty, as it is
based on counts from a single decade. This is particularly relevant given that the historical
simulations do not capture the observed internal variability. To assess whether the model
produces a reasonable number of MSSWs, it would be more informative to construct a
histogram of event counts across multiple decades, both in the simulations and in ERA5. It
would also be useful to examine the seasonal distribution of MSSWs, i.e., the months in
which they tend to occur. In your current plot, based on the selected decade, it appears

**that both models simulate MSSWs earlier in the season compared to ERA5, and the events also seem to be shorter in duration.**

We agree with the reviewer's concerns. A solid classification of the SSWs requires a larger sample size and a more detailed study of both configurations. Both would currently require considerable effort, especially an in-depth analysis of the SSWs of both configurations would increase the scope of this chapter. However, we accept the reviewer's suggestion and plan a separate publication on the Northern Hemisphere stratosphere in ICON XPP. For the presented paper, we therefore removed the two sections related to the polar vortex and SSW, including Figure 18 of the original manuscript.

**Review #2**

"This manuscript describes ICON XPP, a new Earth system model configuration that substitutes the previous MPI Earth system model, as part of the ICON framework. ICON XPP is tailored for climate predictions and projections at two available model resolutions, and aims to be used in the CMIP7 suite of models, after further tuning.

**General review:**

The manuscript provides a valuable contribution to the climate modelling community. The manuscript comprehensively documents and evaluates the ICON XPP model components, configuration and its tuning on decadal and seasonal simulation scales, at both model resolutions. Key drivers of climate prediction and projection are presented and assessed, comparing against other well-established models and ERA5 observational datasets. Assessment and comparison of the models' ability to simulate key measures of climate state, such as climate sensitivity and also key dynamical processes clearly demonstrates where the model performs well and where further improvement could be beneficial. Overall, the manuscript is generally well-written, and the authors effectively present the evaluation of this new Earth system model using key metrics.

I therefore find this paper suitable for publication in GMD, after the author's response/ corrections to a few minor revisions below:

**General:**

- Overall, the manuscript is well-written, however I believe the manuscript could be condensed as there are a number of lengthy sentences/ paragraphs throughout, that should be more concise for clarity and understanding.
  - We agree. As suggested by the reviewer, we shortened the abstract and restructured the introduction. See individual comments below. We also checked the entire text for clarity.
- There are also a number of spelling and grammatical mistakes throughout, I have highlighted some of these below, but I recommend a thorough read through to correct these.

We read the text thoroughly with a focus on spell-check and grammatical mistakes. In particular, several co-authors have made a cross-check with regard to the readability of the manuscript.

**General Abstract:**

The abstract is too long, in my opinion. Ideally the abstract could be condensed into a more concise version, with only key results and information presented. There is some information I believe could be removed/ sentences condensed to reduce the size.
 Agreed. The abstract is reduced by a couple of sentences and subordinate clauses. For example, we have identified the role of the model initiative from which ICON XPP is results, and some very specific text passages about the basic assessment, ENSO and the stratospheric dynamics as clear candidates for cut off.

**Introduction general:**

 While the introduction provides a good presentation of background information required for understanding and guiding the reader through the manuscript, I find the could be structured improved. There is some repetition in some paragraphs, or others seem to continue discussing a point made in previous paragraphs throughout all paragraphs. I would recommend some minor restructuring of the introduction, so it easily flows and provides a succinct background to the great main body of the manuscript.

e.g Line 82: "Here, we present ICON XPP, from the design of the configurations to a first evaluation of the Earth System state based on the CMIP DECK (Diagnostic, Evaluation and Characterization of Klima) experimentation framework (Eyring et al., 2016)."

This reads as if it should be placed at the end of the introduction? But this could be a personal opinion.

Yes, we agree. We sharpened the individual paragraphs by shuffling a couple of sentences. For example, the sentence including "...new modeling initiative..." is placed in the 2nd paragraph, where the model structure is described. Also, the CMIP DECK experiment is now placed in the paragraph about climate projections. The climate projection paragraph is shifted upward, right after the predictions, and fits much clearer the textual order (prediction, projections, research).

In addition, some subordinate clauses have been deleted. On example is – as already mentioned by the reviewer – the relative complex last sentence in the first paragraph, which is rephrased to: "Special attention is given to monitoring certain aspects of the tropical and extra-tropical mean climate, including key modes of variability. "

**Minor corrections:**

- Line 47: "At regional scale". Should this read: "At a regional scale"?
   Changed accordingly.
- Figure 1 and Figure 8: The colours used for each model resolution is switched in these figures, ensuring all figures have a consistent colour scheme would improve clarity.

  Yes, sorry! The colors in now figure 9 is now synchronized with figure 1.
- Lines 76-78: "Since 2020, a new modelling initiative integrating numerical weather forecast, climate predictions and climate projections based on the ICON framework (Müller et al., 2025)." This sentence seems unfinished?

This sentence is rephrased and shifted to the next paragraph. It will read: "Since 2020, a new modeling initiative integrates numerical weather forecast, climate predictions and climate projections based on the ICON framework into a single model system (Müller et al., 2025)." (177ff)

- Line 79: '-,' The comma isn't needed.
  Agreed.
- Line 80: "ICON XPP will be the baseline for next generation climate predictions ..." Should be: "ICON XPP will be the baseline for the next generation of climate predictions...? Changed accordingly.
- Lines 84-87: "Special attention is given to monitoring certain aspects of the tropical and extra-tropical mean climate, and the stratosphere, including key modes of variability and their predictability, such as the El Niño/Southern Oscillation (ENSO), or the North Atlantic Oscillation (NAO)." This sentence is slightly verbose and clunky.

The sentence is changed to: "Special attention is given to monitoring certain aspects of the tropical and extra-tropical mean climate, and key modes of variability." (169-70)

• Line 88: "for the individual components" - Leaves the reader wondering what the individual components are. Could point to a table or list of the components, or leave the "individual components" part out?

This sentence is changed to:" ICON XPP advances the achievements of previous ICON initiatives related to sub-components of the Earth System model ..." (I71ff)

Line 107: Needs to be clearer

The part "continental-scale temperature and precipitation patterns" is deleted.

• Lines 107-108: "and it is recently for machine learning methodologies to assess..." - Should read, "and it has recently been used for machine learning methodologies to assess..."? If not, this sentence should be re-phrased.

The subordinate clause about machine learning is deleted.

 Line 109: "MPI-ESM has been also used for decadal climate" - Should read, "MPI-ESM has also been used for decadal climate".

Changed accordingly.

• Line 172: "TERRA" isn't defined? A definition or acronym description here would be useful. Thanks! TERRA was adopted from the Latin for "earth". This note is now added to where TERRA first appears.

For clarity also explain the role of TERRA in ICON Land, which has not been stated yet:" ICON Land includes the JSBACH land-surface model developed for predecessors of ICON XPP such as MPI-ESM (Reick et al., 2013, 2021), and other land-surface model such as TERRA (from the Latin for "earth"), which is implemented into the operational configuration of ICON NWP" (I143ff)

Line 264: "2 metre" - Should it read "2 meters"?
 Changed accordingly.

• Lines 262-265: "The targets mainly consider the thermodynamic state of the atmosphere - depicted by the top-of-atmosphere radiation balance and global-mean temperature at 2 metre - and the ocean-cryosphere - by the strength of the Atlantic meridional overturning circulation (AMOC) and sea-ice properties" - A long sentence, could split into 2? The whole first paragraph of the 2.3 Tuning section could be re-worded for flow and clarity.

The sentence is rephrased to: "The targets mainly consider the top-of-atmosphere (TOA) radiation balance and global-mean temperature at 2 meters (GMT) - and the strength of the Atlantic meridional overturning circulation (AMOC) and sea-ice properties. " (I246ff)

- Line 276: "@26° N" is this the correct symbol?
   Changed to "at 26° N"
- Line 277: "a small trend remains for the AMOC at the end of the simulation." Should this read 'by the end of simulation", instead of "at the end of simulation"? This would provide

more clarity for me, implying a trend still remains by the end of the simulation? The authors could also elaborate on this trend more? A negative trend/ the AMOC weakens? Thanks! The sentence is changed to: "In 160/40 a small negative trend of the AMOC at the end of the simulation. " (I265-266)

- Line 284: "sediment,." should be "sediment."? No need for extra comma. Changed accordingly.
- Table 1: Could the sea ice parameters have units? i.e Sea ice melting 0.25, is this
  millionkm²? Or is there a reason no units are provided? Some more information could be
  provided on the tuning process for the reader to understand how each value was reached.
  The sea-ice parameters are dimensionless. We added a statement of the units and whether
  they are dimensionless in the figure caption.

In addition, to get the optimized values we performed a series of tailored experiment targeting the climate benchmarks. We included a statement accordingly in the main text to make this a bit clearer:

"A series of tailored pre-industrial control experiments are employed to find the optimal parameterization values. First, a wider range of convection, microphysics and cloud cover parameters are examined to estimate their impacts on the TOA radiation balance and GMT. Then, with the resulting subset of atmospheric and oceanic parameters the ocean-circulation and sea-ice distributions are adjusted. With the optimized parameters a new spin-up is started. The values of the optimized parameter values are shown in Table 1." (1252ff)

• Line 307: "The climate sensitivity is estimated by 1%...", should read "The climate sensitivity is estimated by a 1%..."?

Changed accordingly.

• Line 324: Again, I may have missed it but "AMIP-type", could be defined or the acronym breakdown provided.

We replace "AMIP-type" by "atmosphere only"

Line 328: "...for 160/40 and ~1.7 °C 80/20,...", missing a 'for'? should read "...for 160/40 and ~1.7 °C for 80/20,..."

Thanks. Is changed accordingly.

• Line 330: "The sea-ice reveal reasonable...", missing 'simulations'? Should read: "The sea-ice simulations reveal"?

Thanks. Is changed accordingly.

- Line 330: "peak season"/ "minimum season" could be defined? As the terms "peak season"/ "winter season/ growth season" etc are used interchangeably. Could be useful to provide the months referred to in brackets.
  - We replaced "peak" by "winter" and "minimum" by "summer". "Winter" and "summer" are defined in the figure caption.
- Line 344: "peak magnitudes of ~15-20 Sv at 26° N at 1000m depth,...", should it read "and 1000m depth..."? In addition, should the peak magnitude be a range, or a single number?

Yes, thanks. We changed this sentence to: "For the last 500 years of simulation, the overturning circulations in the Atlantic at 26° N and 1000 m depth show values between 14-17 Sv for 80/20 and 16-19 Sv for 160/40,..." (I337ff)

- Table 3: misaligned data in the observations column/ Denmark Strait row?
   Changed accordingly.
- Line 423: "a small ensemble of three..." I would be interested to know why three were generated? Was this a computational constraint, or something else? An explanation in-text could be useful?

No, unfortunately not. The number is simply a guess to see whether the experiments roughly do similar by changing the initial conditions. Since this is based more on "modeler's experience" rather than "computational constraints" we would like to avoid a discussion in the text.

• Line 437: "The causes are currently unclear, and further investigations are in progress." Is there any literature available yet? Is it possible to cite these further investigations?

We modified this sentence and provide references to support the text, the new lines are:

"Although the causes of the double-ITCZ are currently unclear, some models have modified the clouds microphysics, vertical entrainment rates, convection schemes or the atmospheric energy balance to reduce this feature (e.g., Ma et al., 2023; Ren and Zhou, 2024); however, no generalized modification can be applied to all models" (I451ff)

Ma, X., S. Zhao, H. Zhang, and W. Wang, 2023. The double-ITCZ problem in CMIP6 and the influences of deep convection and model resolution. International Journal of Climatology, 43, 2369-2390, doi: 10.1002/joc.7980.

Ren, Z., and T. Zhou, 2024. Understanding the alleviation of "double-ITCZ" bias in CMIP6 models from the perspective of atmospheric energy balance. *Climate Dynamics*, 62, 6819-6839, doi: 10.1007/s00382-024-07238-7.

- Line 448: "Smaller errors" could be made clearer? "Smaller margin of error"?

  We changed this sentence to: "Smaller root mean squared errors (RMSE) are found for many dynamical and thermodynamical quantities by increasing the resolution from the 160/40 to the 80/20 configuration." (I475ff)
- Table 4: It would be nice to see Table 4 placed nearer to where comparison to observations is mentioned, if possible.

Changed accordingly.

• Line 577: Could define the "cold tongue bias" better, this sentence could also benefit from some citation.

Thanks, we added the following: "The cold tongue bias refers to the excessive cooling along the equatorial Pacific, a common systematic error in climate models (Li & Xie, 2014)." (I593)

- Li, G., & Xie, S.-P. (2014). Tropical biases in CMIP5 multimodel ensemble: The excessive equatorial Pacific cold tongue and double ITCZ problems.
- Line 588: "Nino3.4-related", could you introduce this? A definition or explanation would be useful here.

Thanks, this sentence now starts with: "A regression of SST anomalies to the Nino3.4 index" to make the Nino3.4 region more explicit. Further the Nino3.4 index is defined in the figure caption. (I605ff)

- Line 590: "As many coupled models..." should read "As in many coupled models..."? Changed accordingly.
- Line 691: "Among other..." should read "Among others..."?
   Changed accordingly.
- Line 698: "... and meanwhile...", do you need both?

  In fact, not both are needed. The sentence is changed to: "...and seasonal and decadal prediction skill of the NAO is established ..." (I719)
- Line 716: Full stop missing after citation (Müller et al., 2018)
   Changed accordingly.
- Line 737: "well-behaviour", is this the appropriate word? Are you referring to good performance?

Thanks. Changed to "good performance"

 Line 786: Brackets are needed around the citation (Niemeier et al., 2024), or sentence needs re-phrasing if not.
 Changed accordingly.

• Line 795: "However, several factors – among others volcanic eruptions...", grammatical errors here/ more punctuation needed.

We removed the two sections on the polar vortex and SSWs (including Figure 18 original manuscript), as it would require more effort to elaborate on the SSWs in the necessary depth. In response to the first reviewer, it would be necessary to consider several decades to quantify the necessary variability of the SSWs, as well as a more in-depth analysis beyond the presentation of the time series. Both would require currently considerable effort, especially an in-depth analysis of the SSWs of both configurations would increase the scope of this chapter.

- Line 806: The sentence would read better if it were "winter northern hemisphere" not "hemispheric"? Punctuation is also missing from this sentence: "shows, for example, ...". Please see comment above.
- Line 812: "exhibits comparable variations as in observations", this sentence could be much shorter e.g "comparable to observations".

Please see comment above.